# Dependence of the shape of graphene nanobubbles on trapped substance

H. Ghorbanfekr-Kalashami[1], K.S. Vasu[2,3], R.R. Nair[2,3], François M. Peeters[1] & M. Neek-Amal[3,4,5]

Van der Waals (vdW) interaction between two-dimensional crystals (2D) can trap substances in high pressurized (of order 1 GPa) on nanobubbles. Increasing the adhesion between the 2D crystals further enhances the pressure and can lead to a phase transition of the trapped material. We found that the shape of the nanobubble can depend critically on the properties of the trapped substance. In the absence of any residual strain in the top 2D crystal, flat nanobubbles can be formed by trapped long hydrocarbons (that is, hexadecane). For large nanobubbles with radius 130 nm, our atomic force microscopy measurements show nanobubbles filled with hydrocarbons (water) have a cylindrical symmetry (asymmetric) shape which is in good agreement with our molecular dynamics simulations. This study provides insights into the effects of the specific material and the vdW pressure on the microscopic details of graphene bubbles.

[1] Departement Fysica, Universiteit Antwerpen, Groenenborgerlaan 171, Antwerpen B-2020, Belgium. [2] School of Chemical Engineering and Analytical Science, University of Manchester, Manchester M13 9PL, UK. [3] National Graphene Institute, University of Manchester, Manchester M13 9PL, UK. [4] Shahid Rajaee Teacher Training University, 16875-163, Lavizan Tehran, Iran. [5] Institute for Advanced Technologies, Shahid Rajaee Teacher Training University, 6875-163, Lavizan Tehran, Iran. Correspondence and requests for materials should be addressed to M.N.-A (email: neekamal@srttu.edu).

Graphene is known to be a robust elastic crystal capable of holding mesoscopic volumes of liquids, gases, organic fluids, hydrocarbons and nanocrystals[1–3]. Such graphene nanobubbles can have sizes from 0.37 nm (which is the minimum observed height of a monolayer of atomically flat water adlayer on mica substrate) to a few micron in height and diameter depending on the initial amount of trapped substance[4–6]. Xu et al.[4] used single-layer graphene as an atomically flat coating to visualize the water adlayer islands grown on mica substrates. Using infrared spectroscopy, the pressure within the formed nanobubble was measured to be around 1 GPa at 600 K (ref. 7). This resulted in novel and unpredicted chemical reactions inside the trapped bubbles due to the presence of extremely high pressure[8]: for instance, it was found that a strong confinement effect is observed on the chemistry of CO molecules captured between graphene and a Pt surface[9]. The size, shape and their statistics can be monitored by various techniques such as atomic force microscopy (AFM)[3]. Because of graphene inertness and its capability of withstanding large strains such nanobubbles offer unique opportunities to investigate nano-quantities of materials under extreme conditions.

Using membrane theory, nonlinear plate theory, and the ideal gas model, Yue et al.[10], studied the mechanics of relatively large graphene bubbles. The pressure inside the large graphene nanobubbles ($R > 10$ nm) is predicted to be in the order of MPa,[10–12] which is essentially determined by the elastic properties of the top layer and the interfacial adhesion between top layer and the substrate, $P \propto \frac{\Gamma}{R}$ (ref. 10), where $\Gamma$ is the adhesion energy per unit area and $R$ is the radius of the nanobubble. Recently, for large bubbles ($R > 50$ nm), Khestanova et al.[13] found experimentally a universal scaling law of $h_{max}/R$ where $h_{max}$ is the maximum height and $R$ is the base radius of the bubble filled with hydrocarbons, as measured by AFM[10,13].

Moreover, for a nanobubble filled by ethylene with $R = 4$ nm and height 0.5 nm, it was shown that this results in a pseudo magnetic field of 100 Tesla (ref. 14) which changes fundamentally the electronic spectrum of graphene. The latter is a consequence of the large induced strain in graphene. Furthermore, possible phase transitions in the trapped substance prevent the use of the ideal gas model, that is, $PV = NK_BT$, and therefore the calculation of the internal pressure for small nanobubbles has been remaining a challenge.

Most of the theoretical studies have addressed the deformation and structure of graphene bubbles on top of a substrate[10,13]. Although the effects of pressure on the chemical equilibrium and kinetics is an archaic topic[7,15] recent observations have shown that the Van der Waals (vdW) pressure can induce unusual chemical reactions where several trapped salts or compounds are found to react with water at room temperature, leading to two-dimensional (2D) crystals of their corresponding oxides[8]. These structural transitions and corresponding chemical reaction mechanisms are not well understood yet.

In this paper, we focus on how the material inside the bubble influences the microscopic shape of the bubble. For illustrative purposes, we considered four materials with very different properties. Hydrocarbons are often present as contaminants[10,13]. Several studies exist on confined water[16–19] while ethanol, helium and NaCl have not been considered. By means of equilibrium molecular dynamics (MDs) simulations and AFM measurement, we are able to provide deeper insights into the microscopic details of the graphene nanobubbles and the internal pressure for small nanobubbles. We study bubbles filled by diverse substances such as helium, water, two hydrocarbons (ethanol, hexadecane) and NaCl. The vdW pressure is found to be in the order of GPa depending on the size of the bubble and the interfacial adhesion. Flat nanobubbles can be formed in case of trapped metallic

substances or for large elongated hydrocarbons. Our AFM experiments indicate that in contrast to water bubbles, the hydrocarbon bubbles have round shape with in-plane cylindrical symmetry which is in good agreement with our MD simulations results. Our systematic study provides a deeper understanding of the formation of graphene nanobubbles beyond simple membrane theory that extends its applicability to small nanobubbles and predicts a substance dependent nanobubble shape.

## Results

**van der Waals pressure.** The size of a bubble depends on the number of trapped atoms/molecules and the induced hydrostatic pressure inside the bubble is determined by the adhesion forces between the layers forming the bubble[20]. Using membrane theory for round shape bubbles, the hydrostatic pressure and adhesion energy of the graphene bubbles are respectively given by[10]:

$$P_{hyd} \cong 2.85 \frac{Yh_{max}^3}{R^4}, \quad \Gamma \cong 1.79 \frac{Yh_{max}^4}{R^4}, \quad (1)$$

where $Y$, $h_{max}$ and $R$ are, respectively, the Young's modulus of graphene (340 N m$^{-1}$), the height, and radius of the bubble. For $h_{max} \sim 1$ nm this results into $P_{hyd} \approx 1.6$ GPa ($\Gamma \sim 1$ N m$^{-1}$) and 100 MPa ($\Gamma = 0.1$ N m$^{-1}$) for bubbles with radii $R = 5$ nm and $R = 10$ nm, respectively. The obtained adhesion energy for larger bubbles are in agreement with previous first-principles calculations[21] and our semi-empirical calculations[22]. In fact, the notable elastic properties of monolayer graphene and the strong interfacial adhesion between graphene and the substrate causes the intercalated atoms to be squeezed into extremely small volumes ($R \sim 1$–10 nm) where they can experience a pressure of the order of GPa. Furthermore, nonlinear plate theory modifies equation (1) as follows[10]:

$$P_{hyd} \cong 2.56 \frac{Yh_{max}^3}{R^4} + 64 \frac{\kappa h_{max}}{R^4},$$

$$\Gamma \cong \frac{Yh_{max}^4}{R^4} + 32 \frac{\kappa h_{max}^2}{R^4}. \quad (2)$$

The second terms in equation (2) are the contributions of bending energy where $\kappa \cong 0.24$ nN-nm is the bending stiffness of graphene. For $h_{max} \sim 1$ nm this gives $P \sim 1.42$ GPa ($\Gamma = 0.56$ N m$^{-1}$) and 89 MPa ($\Gamma = 0.035$ N m$^{-1}$) for $R = 5$ nm and $R = 10$ nm, respectively. It is seen that by including the second terms, pressure decreases with $\sim 10\%$ and the adhesion energy with 50–80%. Notice that in equations (1) and (2) both $Y$ and $\kappa$ varies slightly with temperature[23]. It is important to note that for larger $h_{max}/R$ the contribution of the second terms in equation (2) become more important[10]—see Supplementary Note 1.

In common MDs simulations[20,24], the pressure is calculated using the virial method which requires a homogeneous system[10,11]:

$$P_{hyd} = \frac{1}{V_b} NK_BT + \frac{1}{3V_b} \sum_i <\mathbf{r}_i.\mathbf{f}_i>, \quad (3)$$

where the first term is the ideal gas pressure and the second term is due to the interatomic potential force ($\mathbf{f}_i$). In equation (3), the volume of trapped substance is taken as independent of temperature. Also, note that the rigid boundaries in common simulations cause inhomogeneous and non-equilibrium conditions which may invalidate equation (3).

The above drawbacks are overcome using the stress tensor-based method[25]: for a fluid in equilibrium, the trace of the stress tensor per volume is balanced by the hydrostatic pressure, that is, $P_{hyd} = P_{vdW}$. Therefore, one can use the

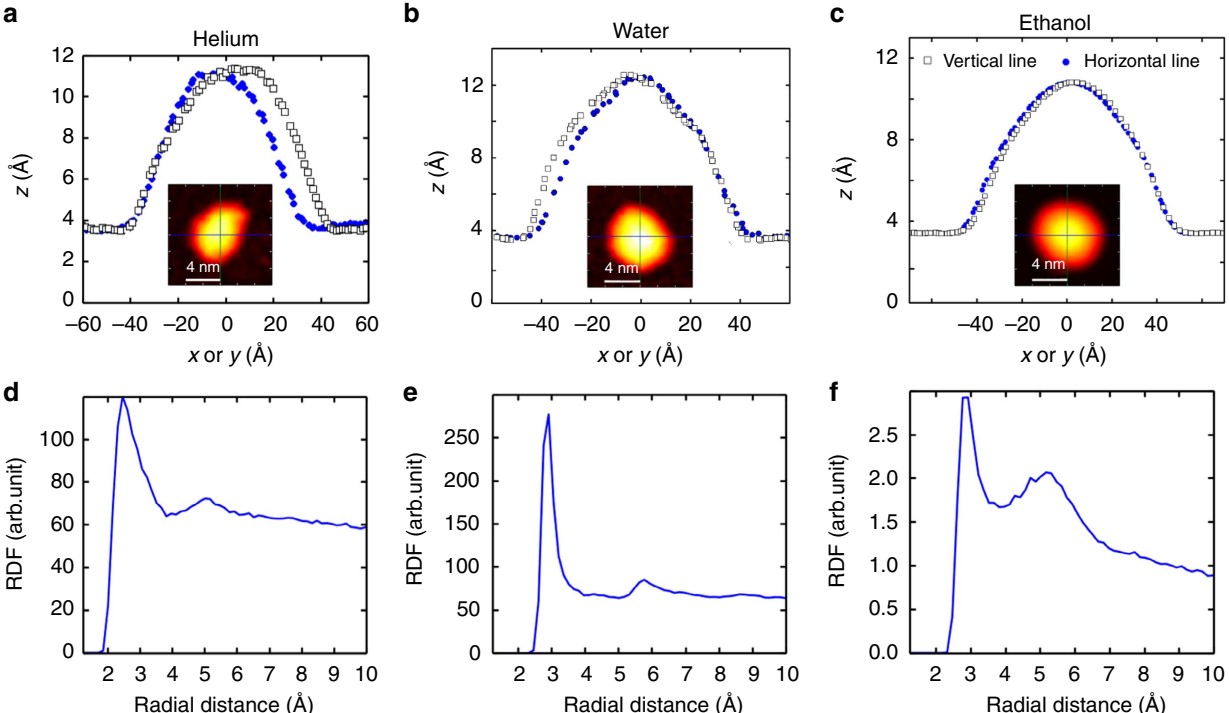

**Figure 1 | Nanobubbles.** The cross-sections of the graphene nanobubbles filled with (**a**) helium (**b**) water (**c**) ethanol at room temperature. The insets represent the height deformation (which is time dependent). (**d**) The corresponding RDF of He–He distance of trapped helium and O–O distance of trapped (**e**) water and (**f**) ethanol.

following equations to evaluate the density ($\rho$) and the vdW pressure ($P_{vdW}$) induced inside a bubble

$$P_{vdW} = -<\frac{Tr(\sigma)}{3V_b}>, \qquad (4)$$

$$\rho = <\frac{Nm_u}{N_A V_b}>, \qquad (5)$$

where $<.>$ represents a time average over several realizations and $\sigma$, $V_b$, $N$, $m_u$ and $N_A$ are, respectively, the trapped atoms stress tensor in unit of pressure × volume[25], bump volume, number of trapped atoms, atomic mass, and Avogadro's number. This method will enable us to find the vdW pressure inside nanobubbles that are filled by inhomogeneous substances with different density—see Supplementary Note 2.

We emphasized that our method based on the virial stress tensor is more general than equations (1) and (2). The latter originates from elasticity theory with the assumption of a round shape and using the large bubble limit. Accordingly, it in general fails to describe various bubble shapes such as semi-circular or non-circular in our simulated system.

**Helium bubble.** First, we simulate 792 helium atoms below the bumped graphene and found the optimized structure at $T = 0$ K. This is equivalent to previous approach based on elasticity theory[13], where the bump energy was minimized with respect to $h_{max}$ and $R$. Subsequently, we increased $T$ up to room temperature, and investigated the spatial structure of the trapped helium atoms. We found that the non-zero temperature results are different from the ground state results (that is, $T = 0$). At room temperature, thermal fluctuations occur in the trapped substance while the bottom graphene layer is supported by the substrate and therefore remained fixed. The bubble is found to fluctuate randomly and has in general a non-round shape. The inset of Fig. 1a shows a typical 2D-plot of

the bubble filled by helium and the height profiles taken along the two cross-sections are shown in Fig. 1a. This figure is a single snapshot of the bubble at a particular time. Based on our MD simulations, we concluded that He forms a non-circular bubble.

Using lateral radial distribution function (RDF) along the direction perpendicular to the substrate ($z$ axis)[19] we found a second peak, see Fig. 1d, indicating that the trapped helium has long range ordering at room temperature and it exhibits a denser phase than its gas phase. Our results are comparable with the experimentally obtained scanning tunnelling microscopy image for graphene nanobubble filled by argon[26]. We do not expect differences between He and Ar. We found that trapped helium behaves like a gel and is highly fluidic. The structure of the bump continuously changes due to thermal fluctuations and the helium bubble diffuses randomly over the graphene substrate.

To obtain further insight in the arrangement of the He atoms, we evaluated the density profile of trapped helium along the $z$ axis (see Fig. 2a). At 0 K, three peaks are observed at 2.75, 4.85 and 7.25 Å indicating a well defined layered structure (that is, solid phase of helium). The inset shows the RDF of each layer at 0 K indicating the same crystal structure in each layer. At room temperature, there is only a single peak (2.70 Å) representative for a single He wetting layer close to the bottom graphene layer and a disordered arrangement of He atoms above it.

It is worthwhile to mention that for a He bubble at 0 K, we found a layered structure and the corresponding microscopic structure confirms the well known hexagonal close packed (h.c.p.) lattice structure of helium at 0 K which is consistent with the results reported by Hodgdon et al.[26,27].

Using equations (4,5) the density of trapped helium at room temperature and corresponding induced pressure are estimated to be $0.225$ g cm$^{-3}$ and $0.65$ GPa, respectively. Notice equation (2) overestimates the pressure, that is, using $h_{max}$ and $R$ given in Table 1, $P_{hyd} = 1.25$ GPa. The latter difference is due to the non-round shape of the helium bubble. The obtained fluid helium

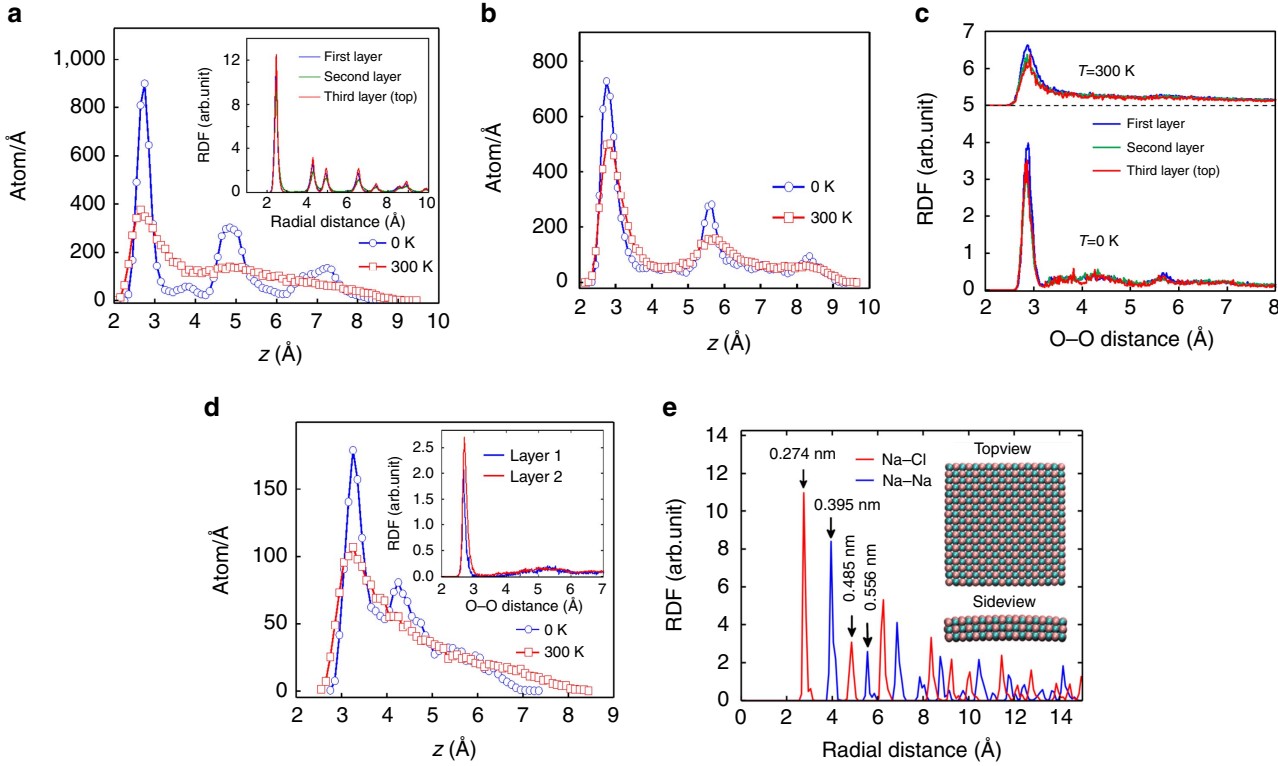

**Figure 2 | Density profiles and RDFs.** (**a**) The density of the trapped helium along $z$ direction for 0 and 300 K. The inset represents RDF of each individual layer of trapped helium at $T=0$ from bottom to top. (**b**) Density profile of the oxygen atoms of trapped water along the $z$ axis at zero and room temperature. (**c**) Corresponding RDF of O–O distance for the three layers of water at 0 and 300 K. The RDF at room temperature is shifted by 5 Å. (**d**) Density profile of the oxygen atoms from trapped ethanol for zero and room temperature. The inset shows the oxygen radial distribution for the first two layers at $T=0$. (**e**) MD predicted partial Na–Na and Na–Cl RDF of encapsulated square NaCl at 0 K. The inset shows corresponding side and top views of the bubble. The simulated graphene top and bottom flake are not shown.

**Table 1 | The geometrical and physical properties of bubbles filled with three investigated substances with corresponding bulk cases shown for comparison.**

| Trapped substance | $R$ (nm) | $h_{max}$ (nm) | $h_{max}/R$ | $P_{vdW}$ (GPa) bubble | $\rho$ (g cm$^{-3}$) bubble | $\rho$ (g cm$^{-3}$) bulk | $\rho$ (g cm$^{-3}$) bulk, $P=1$ atm |
|---|---|---|---|---|---|---|---|
| Helium | 4.5 | 0.83 | 0.18 | 0.65 | 0.225 | 0.283 ($P=0.65$ GPa) | 0.004 |
| Water | 4.6 | 0.90 | 0.20 | 0.93 | 1.103 | 1.191 ($P=0.93$ GPa) | 1.044 |
| Ethanol | 5.4 | 0.78 | 0.14 | 0.49 | 0.987 | 1.051 ($P=0.49$ GPa) | 0.862 |

density is comparable with experimental results (0.28 g cm$^{-3}$) where a nonlinear increase of the fluid helium density was measured up to a pressure of 2 GPa at room temperature[28]. We found that the pressure within the nanobubble increases with temperature due to the increasing kinetic energy contribution to the stress[25]. The latter is consistent with recently reported results using a hydrothermal anvil cell made of graphene nanobubbles on diamond[5] and with the high-pressure chemistry in the graphene bubble by monitoring the conformational change of pressure-sensitive molecules[7].

**Water bubble**. Water is a polar liquid with a hydrogen bond network. We theoretically predicted previously the square-rhombic lattice structure for monolayer ice confined between graphene layers[17]. Here, we trapped 792 water molecules below the bump and report room temperature results. We found that the water bubble is immobile even when temperature fluctuations

are present. The fluidity of water under the graphene flake is less than that of helium which can be seen from the corresponding movies—see Supplementary Movies 1 and 2. A typical 2D-plot of the deformation of graphene is shown in the inset of Fig. 1b. The corresponding profiles along the $x$- and $y$-directions and the RDF are given in Fig. 1b,e, respectively. The first peak in Fig. 1e corresponds to an O–O distance of 2.8 Å which is consistent with results from neutron diffraction spectroscopy[18]. Notice that there is a second peak in the RDF, and therefore water has weak long range ordering and exhibits a rather amorphous structure.

Next, we considered larger bubbles and compare them with typical bubbles measured in experiment. In Fig. 3a,b, we depict an AFM image of a water bubble and corresponding profiles along two perpendicular lines. We performed extensive simulations for larger water bubbles having average in-plane radius of 18 nm. The result is presented in Fig. 3c and corresponding profiles along two perpendicular lines are shown in Fig. 3d. Although the size of the experimental bubble is larger than our simulated sample, both

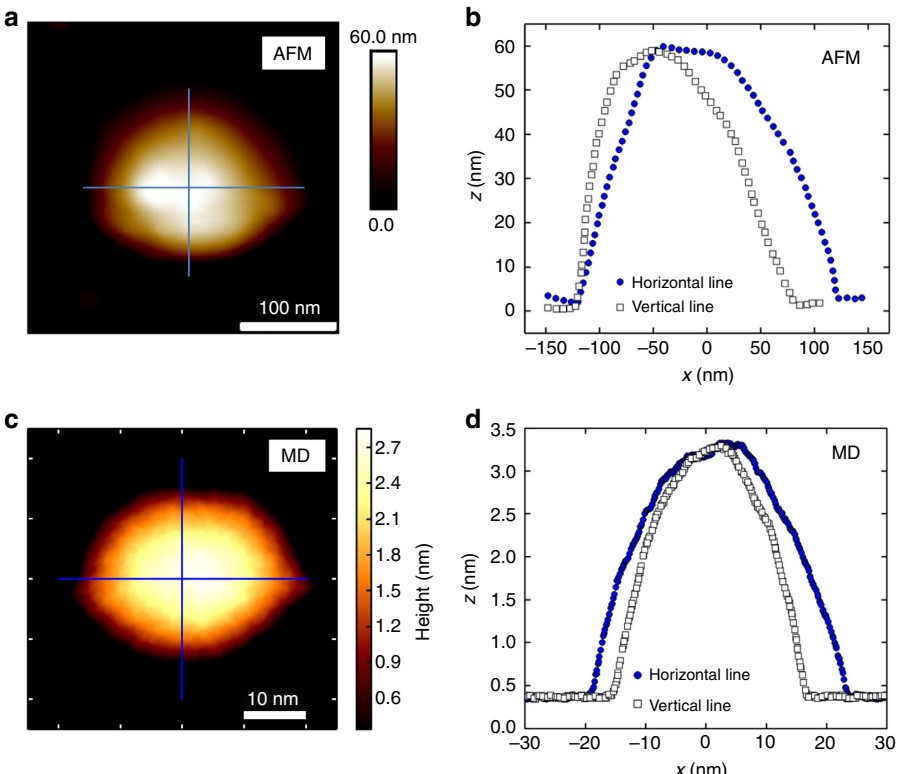

**Figure 3 | AFM measurement and MD simulation of large water bubble.** (**a**) AFM image of water bubble and (**b**) the corresponding profiles along two lines. (**c**) Our MD results for water bubble having $R = 18$ nm and (**d**) two corresponding profiles along two perpendicular directions.

AFM image and MD results confirm the non-round shape of the water bubble. More AFM images for water bubbles are presented in Supplementary Note 3 and Supplementary Fig. 1a,b.

Figure 2b shows the density profile of the oxygen atoms along the $z$ direction at two different temperatures 0 and 300 K. At 0 K, the trapped water molecules have a layered structure, that is, it shows three layers which are located at $z = 2.76$, 5.66 and 8.36 Å. Surprisingly, confined water preserves its layered structure even at room temperature (representing the amorphous-solid-like phase for trapped water). The corresponding peaks are located at 2.82, 5.76 and 8.65 Å. At higher temperature, the inter-larger distance becomes larger. The density and the induced vdW pressure are estimated to be $\rho = 1.03$ g cm$^{-3}$ and $P_{vdW} = 0.93$ GPa. Again, due to the non-round shaped water bubble equation (2) overestimates the pressure, that is, $P_{hyd} = 1.4$ GPa. The present study on the size effect shows the limitation of equation (2)—see section Size effects.

A wide range of stable amorphous ice structures are possible due to the adaptability of perturbations in hydrogen bonded networks between the water molecules. All water layers exhibit a similar microscopic structure at room temperature, which can be seen from Fig. 2c where we show the RDFs of the O–O distance for each individual layer for both 0 and 300 K. In Fig. 2c, notice that the water bubble at room temperature is less ordered as compared to the 0 K one. A square-rhombic structure is found only at the bottom layer of confined water where $T = 0$ K. However, the top layers do not show any signature of the square ice structure and are less ordered than the bottom layer. In contrast to ref. 24, we argue that the density and size of water bubbles are important factors that control the microscopic structure of trapped water which was not considered in other theoretical work[16]. In the latter study, confined water between two parallel graphene sheets was studied for different interlayer distances (6–12 Å) using MD simulations. At high pressure

($\sim 1$ GPa), they observed AA stacking of two layers of square ice while in our simulated bubble we found AB stacking as in ref. 17.

**Hydrocarbon bubble: ethanol and hexadecane.** It is noteworthy to look at the microscopic structure of trapped hydrocarbons. We found very good agreement between our MD results for bubbles filled with small hydrocarbons, the predictions from elasticity theory (equations (1) and (2)), and AFM measurements. As an example, we simulated 200 trapped ethanol molecules. At room temperature, the average shape was found to be circular. In Fig. 1c, we show a 2D-plot of the bump (inset) and its profile across indicated lines. The RDF of O–O shown in Fig. 1f indicates that trapped ethanol is in the liquid phase. The estimated density and pressure are $\rho = 0.987$ g cm$^{-3}$ and $P_{vdW} = 0.49$ GPa, respectively. This is consistent with the fact that ethanol crystallizes for pressures around 1.5 GPa (ref. 29). Using the numbers given in Table 1 and equation (2) gives $P_{hyd} = 0.5$ GPa, which is in very good agreement with our MD results that is due to the round shape of the bubble. Therefore, the experimental nanobubbles reported in ref. 13 are more likely to be filled by hydrocarbon (ethanol, methanol and other small hydrocarbons) contaminants rather than water or rare gases. Our results are qualitatively in agreement with the experimentally reported liquid phase of bulk ethanol under 1 GPa pressure from dielectric spectroscopy[29].

In Fig. 4a,b, we depict an AFM image of a hydrocarbon filled bubble (likely filled by small hydrocarbons) and corresponding profiles along two perpendicular lines. Raman spectroscopy as characterization technique confirms the presence of trapped ethanol (more details are available in Supplementary Note 4 and Supplementary Fig. 3a,b. In Fig. 4c,d the MD results for a larger ethanol bubble is shown ($R \sim 15$ nm). Again, the size of the bubble is larger than our simulated sample, however the AFM

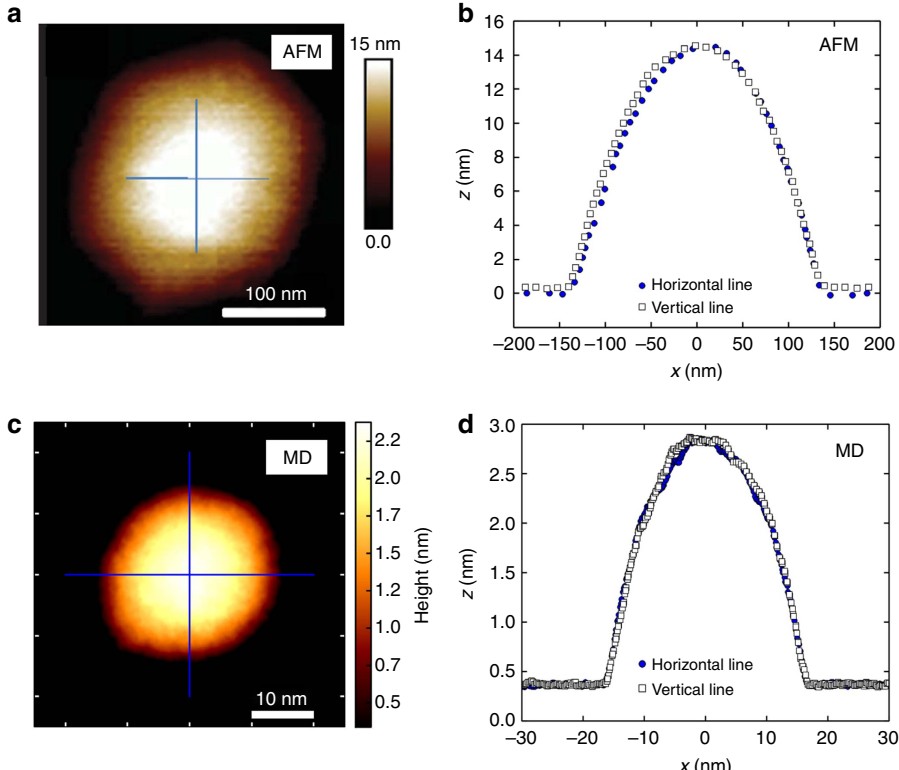

**Figure 4 | AFM measurement and MD simulation of large ethanol bubble.** (**a**) AFM image of a hydrocarbon bubble and (**b**) the corresponding profiles along two lines. (**c**) The MD results for larger ethanol bubble having size of 15 nm and (**d**) two corresponding profiles along two perpendicular lines.

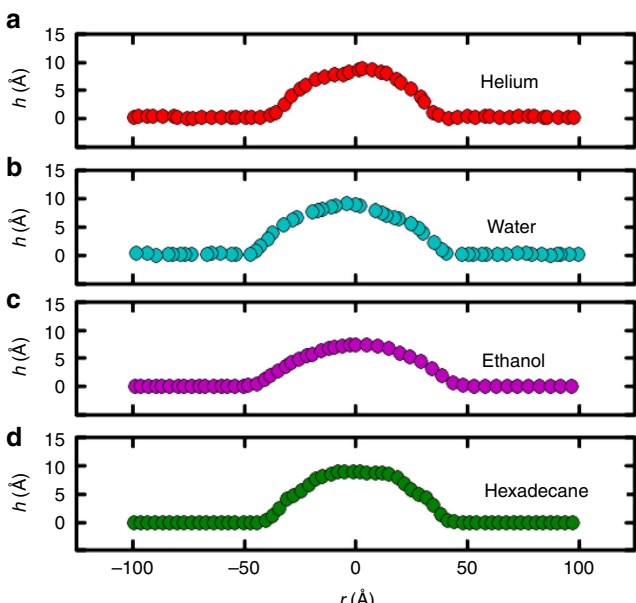

**Figure 5 | Height profile of different types of bubbles.** MD predicted height profile for helium (**a**), (**b**) ethanol, (**c**) water and (**d**) hexadecane (long hydrocarbon) at room temperature. The latter forms a bubble with larger flat area at the top of the bubble in contrast to the other substances.

images show a round shape with in-plane cylindrical symmetry in agreement with our MD results. More AFM images for hydrocarbon bubbles are presented in Supplementary Fig. 2a–d and Supplementary Note 3.

In Fig. 2d, we show the density profile of the oxygen atoms of trapped ethanol for both zero and room temperature. Peaks at 3.4, 4.2 and 5.6 Å indicate that at 0 K trapped ethanol has a layered structure. The inset shows similar pronounced peaks of the RDF of the two first layers indicating that they have similar ordering. As temperature increases, only a single wetting layer next to the substrate is formed[29].

Next, we performed additional simulations using large hydrocarbons, that is, $C_{16}H_{34}$ known as hexadecane. We captured 160 molecules and optimized the structure. We found that they tend to be aligned with each other and form a bubble with a larger flat area. In Fig. 5, we compared the MD predicted height profile of helium, ethanol, water and hexadecane (long hydrocarbon) at room temperature. In our MDs simulations without residual strain in graphene, small hydrocarbon molecules (ethanol) always form round shape bubbles with in-plane cylindrical symmetry and water forms non-round shape bubbles while large linear molecules are arranged in a crystalline-like structure (that is, being aligned) and form a bubble. There is a flat region on top of the hexadecane bubble.

In Table 1, we report geometrical properties such as the bump radius ($R$), maximum height ($h_{max}$), and aspect ratio ($h_{max}/R$) for the studied small nanosize bubbles ($R < 10$ nm). We obtained $h_{max}/R$ values between 0.17 and 0.20 which is larger than the experimentally reported value of 0.11 for large bubbles ($R > 10$ nm) and it is in agreement with the experimental results on small size bubbles[13]. Therefore, in agreement with the experimental results of ref. 13 the universal scaling (that is, $h_{max}/R \cong 0.1$) is not applicable for small size bubbles ($R < 5$ nm).

We compared the density of trapped materials with those for bulk helium, water, and ethanol at room temperature at two different pressures and listed the results in Table 1. The density of encapsulated ethanol is larger than that for bulk at normal condition. This is additional support for the high vdW pressures

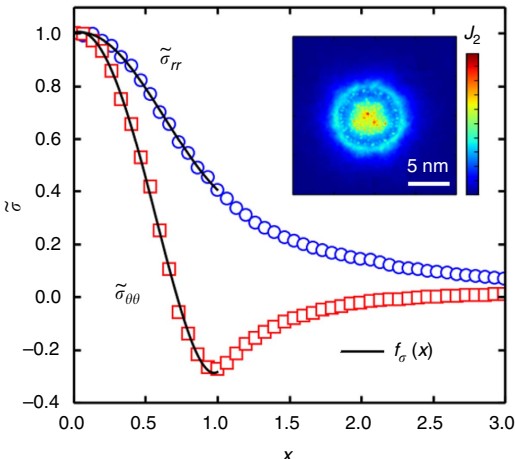

**Figure 6 | Stress of the top graphene layers when the bubble is filled with ethanol.** MD predicted radial ($\tilde{\sigma}_{rr}$) and circumferential ($\tilde{\sigma}_{\theta\theta}$) stress components of an optimized ethanol bubble ($R \sim 4$ nm) as a function of normalized radial distance ($x$) at 0 K. The results can be fitted by a polynomial function $f_\sigma(x)$ within the region of bubble ($x < 1$). Inset shows the corresponding stress distribution based on the value of $J_2$ (equation (8)).

present inside the nanosize graphene bubble. In Supplementary Note 5, we studied the deformation profiles of the graphene sheet—see Supplementary Fig. 4a–c. For small bubbles, it is found that it can be fitted by a polynomial function which does not necessarily follows the universal scaling law[13].

**Stress calculations.** Here, we study the radial and circumferential components of the stress tensor ($\sigma_{rr}, \sigma_{\theta\theta}$) of the top graphene sheet for ethanol bubbles and found very good agreement with those predicted by membrane theory[10]. In Fig. 6, we present the average $\sigma_{rr}$ and $\sigma_{\theta\theta}$ as a function of radius obtained from our MD simulation for an ethanol bubble (radius $\sim 4$ nm) at 0 K. Stress components and radii are scaled to the maximum stress and bubble radius ($R$), respectively:

$$\tilde{\sigma}_{rr} = \sigma_{rr}/\sigma_{rr}^{max}, \quad \tilde{\sigma}_{\theta\theta} = \sigma_{\theta\theta}/\sigma_{\theta\theta}^{max}, \quad x = r/R \qquad (6)$$

The solid back curves in Fig. 6 are prediction from elasticity theory—see Supplementary Note 1. We found that only for ethanol elasticity theory provides good fits of the MD data. Using the nonlinear plate model[10], the $\sigma_{rr}$ and $\sigma_{\theta\theta}$ stress components can be determined from $f_\sigma(x)$ which is a polynomial function of the radius ($x < 1$) given by

$$f_\sigma(x) = 1 + Ax + Bx^2 + Cx^4 \qquad (7)$$

The quality factor defined as $Q = 1 - \sqrt{\frac{1}{N}\sum_{i=1}^{N}(\sigma_i - f_\sigma(x_i))^2}$ for two sets of fitting parameters of $f_{\sigma rr}(x)$ and $f_{\sigma\theta\theta}(x)$ results in the values of 0.994 and 0.987, respectively (see Supplementary Table 1). The inset of Fig. 6 shows the stress distribution of the carbon atoms of the top graphene sheet for the ethanol bubble at 0 K. The colouring is based on the value of $J_2$ (ref. 30) which is determined by

$$J_2 = \frac{1}{6}\left[(\sigma_{xx} - \sigma_{yy})^2 + (\sigma_{xx} - \sigma_{zz})^2 + (\sigma_{yy} - \sigma_{zz})^2 + 6\left(\sigma_{xy}^2 + \sigma_{xz}^2 + \sigma_{yz}^2\right)\right]. \qquad (8)$$

**Salt (NaCl) bubble.** As an example of a very different material, we simulated 1,000 NaCl molecules with four different initial nanocrystal shapes, that is, round, triangular, ellipsoid, and square. We found that the encapsulated NaCl nanocrystals keep

their cubic structure. This may be relevant to ref. 20 that proposed that the reported experimental data on square ice[16] can be better explained by NaCl contaminants that are precipitated as nanocrystals in the dried-out graphene liquid cells. The deformation of the top graphene flake with four different initial shapes (round, triangular, ellipsoidal and square) are shown in Fig. 7a. The crystal will adapt the shape of graphene by forming rough surfaces. In all cases optimized shapes remain unchanged even under high vdW pressure between the graphene cover and the substrate. This is in contrast to the minimized configuration for trapped helium, water and hydrocarbons, regardless of their initial configurations of the molecules result always in semi-circular bump shapes. The corresponding height profile along horizontal and vertical lines, as indicated in Fig. 7a, are respectively shown in Fig. 7b,c. We found that by increasing the size of the initial NaCl crystal (which contains larger flat side) in the x–y plane, the flat region in the formed bubble increases. Such flat bubbles were found in ref. 16 further supporting the proposal made in ref. 20. Although the resistance of the NaCl crystal against the lateral pressure induced by the graphene flake might be as expected, our calculations, reveal the atomistic details for this phenomena.

In Fig. 7d, we depict the density profile of the simulated square shaped NaCl bump along the z direction. The height of the peaks and their distance ($\sim 2.9$ Å) correspond to the cubic crystalline structure of NaCl which is not influenced by the vdW pressure in the nanocapillary. The inset of Fig. 2e depicts side and top views of the minimized encapsulated sample of square NaCl at 0 K. We calculated the partial RDF of the Na–Na and Na–Cl distance and show it in Fig. 2e. The first peak appears at 2.74 Å which is very close to the reported structure in ref. 20, that is, 2.8 Å. The presence of many peaks is an indication of the crystalline structure of encapsulated NaCl which can be accurately approximated by its original cubic-structure. By increasing temperature up to room temperature we found almost the same RDF indicating that the solid phase of NaCl is preserved at room temperature (results are not shown here). We attribute this to the large bulk modulus (24.42 GPa), Young's modulus ($\sim 40$ GPa) and shear modulus ($\sim 12.6$ GPa) of the NaCl crystal[31,32] which is larger than the pressure inside the nanocapillary (1–2 GPa). The higher elastic modulus of the NaCl crystal results in the final cubic structure. The latter originates in the strong ionic bond between Na$^+$ and Cl$^-$ which is $\sim 1,000$ kJ mol$^{-1}$, two orders of magnitude larger than the hydrogen bond, that is, 10 kJ mol$^{-1}$ in ice.

**Layered structure.** The aim of this section is to provide a closer look into the structure of the trapped materials inside the bubble and elaborate on their layered structure. It is important to note that different bubbles (helium, water and ethanol), regardless of their size and/or type, have perfectly optimized round shape at zero temperature. Even at higher temperature as the bubble gets smaller, it attains a more round-like shape. This is the reason why we obtained a non-circular shape for the large water bubble in agreement with AFM experiment (Fig. 3), while it is semi-circular for small bubble size (see Fig. 5). As an extreme example, we do not expect non-round shape of a water bubble filled with too few number of water molecules due to the lack of sufficient H-bonds. Figure 8a presents the RDF of He–He atoms for bulk and trapped helium at room temperature. The two RDFs are very close together which supports the presence of pressures of order GPa inside the nanosize helium bubble. Different views of the layered structure of trapped helium at zero temperature are presented in the inset of Fig. 8a. For larger helium bubble, our MD simulation results in the h.c.p. structure of encapsulated helium. However, at

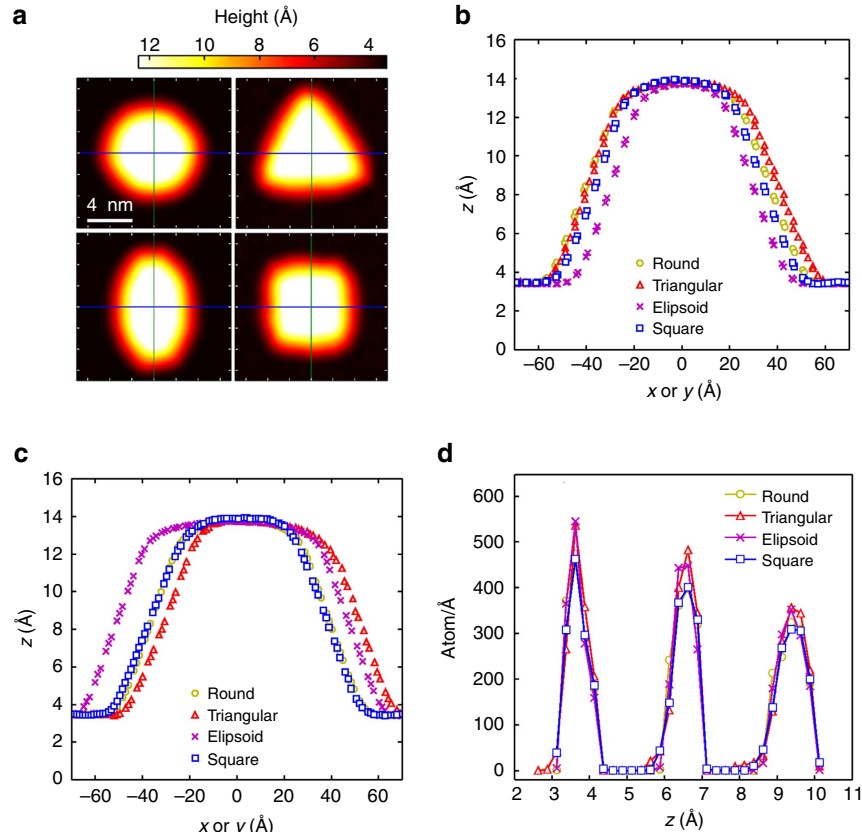

**Figure 7 | NaCl bubbles.** (**a**) 2D-plot of height profiles of graphene nanobubbles filled with NaCl for four different initial configurations: round, triangular, ellipsoid and square shape at $T = 0 K$. The height profile of corresponding cross-sections along indicated horizontal (**b**) and vertical (**c**) lines in **a**. (**d**) The density profile of Na/Cl atoms along $z$ axis for the four NaCl bumps.

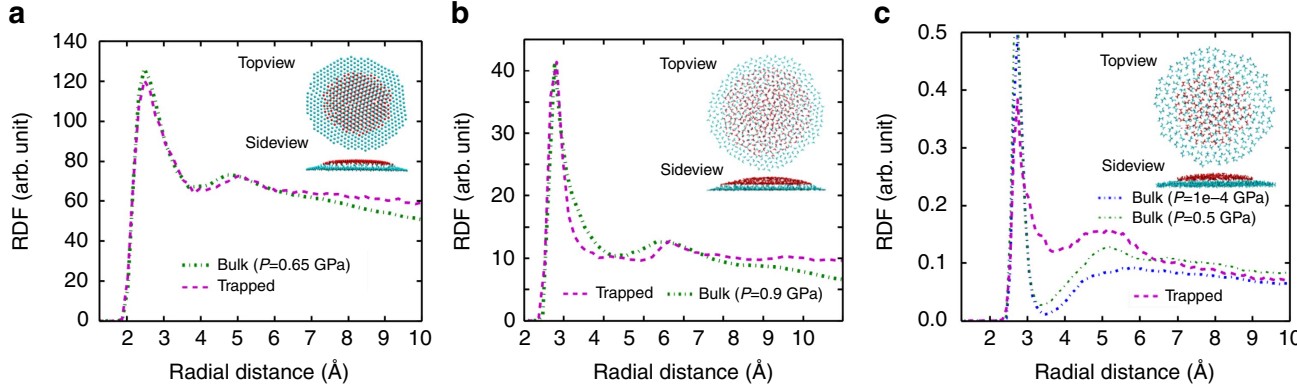

**Figure 8 | Different types of bubbles and comparing with bulks.** The RDF of trapped and bulk (**a**) helium (**b**) water (**c**) ethanol subjected to the same amount of pressure at room temperature (see Table 1). Each inset shows top and side views of corresponding trapped substances. The cyan and red balls represent the two different layers (graphene top layer and the graphene substrate are not shown here).

higher temperature, such a h.c.p. lattice and the layered structure no longer exist (see Fig. 2a where different colours indicate different trapped material layers).

Fig. 8b shows RDF of bulk and trapped water subjected to the same pressure 0.9 GPa at 300 K (see Table 1). The presence of the second peak and the resemblance of the two RDFs indicate that the entire structure of water inside the bubble does not resemble a solid phase. The top and side views of the formed layers at 0 K, shown in the inset of Fig. 8b, clearly demonstrate the layered structure of water. Similarly, the first and the second layer

are indicated by different colours. It can be seen from the figure that the bottom layer of water has likely an ordered structure due to the high confinement, however upper layers have an amorphous structure. At low temperature, our simulation predicts that water bubble have a more round shape.

Finally, Fig. 8c shows the RDF of O-O atoms for bulk and trapped ethanol for two values of pressure (1 atm and 0.5 GPa) at room temperature. For bulk, the second peak becomes pronounced when we increase pressure (see Table 1). This indicates that encapsulation induces high pressure inside the

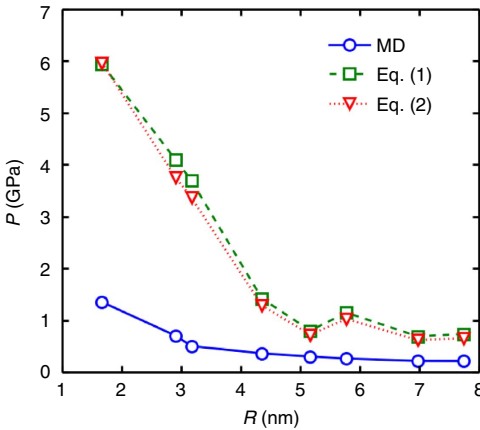

**Figure 9 | Bubble size and induced pressure inside the bubble.** Calculated pressure by MD (equation (4)) and elasticity theory for ethanol bubble as a function of radius. Equations (1) and (2) are from the membrane and the nonlinear plate models, respectively.

ethanol bubble. The top and side views of the layered structure of trapped ethanol at 0 K are shown in the inset of Fig. 8c. The first and the second layer are plotted in different colours. The ethanol bubble has a perfect round shape with distinguishable layers. The latter is due to the interaction with the substrate.

**Size effects.** The size of the bubble which is determined by the amount of trapped material has an impact on the physical properties of the bubble such as the density, the induced pressure, and the layered structure. To investigate this effect, we carried out different simulations in order to compare $P_{vdW}$ and $\rho$ obtained from MD and elasticity theory. Figure 9 shows the calculated pressure versus the radius of the ethanol bubble. The predicted pressure is calculated through the stress tensor (equation (4)) and elasticity theory (equations (1,2)). As can be seen, for small radius ($R < 4$ nm) the latter overestimates the pressure inside the bubble while for larger bubble radius the two approaches result in the same pressure. We believe that our simulated bubble would converge to the predicted pressure as obtained from elasticity theory in the limit of a large bubble. However, for a small bubble both membrane and nonlinear plate models predict unrealistic extremely high pressures. Note that barely noticeable difference between the two models is due to the small contribution of the bending rigidity ($\kappa$) to the elastic energy of graphene beyond the length scale of 4 Å —see Supplementary Note 6 and Supplementary Fig. 5a,b.

### Discussion

We presented an extensive atomistic MDs simulation study to probe the vdW pressure at atomically smooth graphene interfaces by intentional trapping different types of materials (water, gas, hydrocarbons, and salt). The correlation between the shape of the graphene bubble and the physical properties of the trapped substances was pointed out. Temperature and concentration of trapped material have strong influence on the bubble formation. Our detailed simulations along with the experimental observations provides insightful information about the formation of bubbles (including flat nanobubbles) and the effect of vdW pressure on the structural and conformational changes of the trapped substance.

We found that the bump shape and height depend on the thickness and elastic properties as well as on the specific trapped molecules. Experimental investigations on the geometrical and physical properties of graphene bubbles filled with different

substances have motivated us to perform extensive atomistic simulations to determine the microscopic structure of the trapped substances and the corresponding deformation of the graphene flake. We tested five different substances namely, helium, water, ethanol, hexadecane and NaCl inside the graphene nanobubble with an effective radius smaller than 10 nm using appropriate interatomic potentials. Significant differences in the microscopic details of the bubbles were found for different encapsulated substances. Trapped helium behaves like a gel with the highest degree of fluidity subjected to a pressure of 0.65 GPa, while NaCl is very rigid and resists against pressures of order GPa and forms a flat like bubble in contrast to the bubbles found for helium, water and ethanol. On the other hand, helium exhibits a layered structure only at low temperature while water and ethanol preserve their layered structure even at room temperature. The deformed graphene over NaCl follows the shape of the NaCl crystal, except if it in solution. An amorphous and layered structure of water was found even when subjected to 0.93 GPa pressure at room temperature. The water nanobubbles do not have a perfect round shape. Ethanol behaves like a liquid at room temperature, while it forms an amorphous solid at 0 K. The deformation profile for bubbles filled by hydrocarbons (ethanol) are more relevant to our AFM images and the recently reported graphene nanobubbles[13] and the corresponding pressures can be obtained using elasticity theory. Long hydrocarbons, that is, hexadecane, form a more flat like bubble at 0 and 300 K. The pressure inside the bubble can be tuned by changing the adhesion energy which results in an ordered phase of the trapped substance (that is, amorphous-ice in case of water).

In Supplementary Note 7, we found that boundary stress on the graphene flake results in the formation of wrinkles and removes the round shape of the water nanobubbles—see Supplementary Fig. 6a,b. By observing the shape of the bubble it should be possible to obtain information about the physical and chemical changes that may occur in nano-enclosures under the influence of high pressure and different temperatures[33,34]. As mentioned in a previous report[35], Raman spectroscopic measurements on nanobubbles have clearly shown that a small amount of strain develops in the graphene flake depending on the height of the bubbles. For example, the strain developed in graphene for a bubble of height 60 nm is ~0.7.

Moreover, we have shown that even by increasing the adhesion energy between two graphene sheets, which is equivalent to increasing the vdW pressure, a more ordered structure for water is obtained, while the square ice structure and its stacking structure cannot be seen (see Supplementary Note 8 and Supplementary Fig. 7). We found that the bubbles filled with small hydrocarbons can be reasonably well described by elasticity theory in contrast to the other investigated material materials and are therefore suitable test materials for studies of elasticity theory. Furthermore, there are also various accurate force fields available for hydrocarbons which allow to simulate them accurately using MD simulations. The microscopic structure of trapped water and ethanol are very different, for example, the RDF of bulk ethanol and trapped ethanol are very different than that of water. The latter is due to the different distribution of H-bonds in ethanol. Finally, we concluded that previously proposed universal scaling law of height and radius of bubbles is limited to round shape bubbles such as found for graphene bubbles filled with hydrocarbons. But, generally such scaling law is not applicable for any bubble type. As an extreme case, NaCl filled bubble does not follow the universal scaling law found in ref. 13. Our study provides fundamental insights into the formation of graphene nanobubbles and the effect of the microscopic details of the trapped substances in it.

## Methods

**MD simulations.** We used MD simulations using reactive force field ReaxFF[36] to simulate the interaction between carbon, oxygen, and hydrogen in case of water bubble. A Lennard–Jones potential is used to describe the helium-helium ($\varepsilon_{He} = 0.02166$ kcal mol$^{-1}$, $\sigma_{He} = 2.64$ Å) as well as the helium–carbon ($\varepsilon_{He-C} = 0.0334$ kcal mol$^{-1}$, $\sigma_{He-C} = 2.98$ Å) interactions[28]. In the present work, all simulations were carried out using the large-scale atomic/molecular massively parallel simulator[37].

To simulate trapped ethanol, we used the hybrid optimized potential for liquid simulations (OPLS) potential[38] for ethanol and ReaxFF potential for the graphene layers. Liquid and solid ethanol was simulated by using the molecular model proposed by Jorgensen et al.[38]. In a recent study, the introduced OPLS potential includes different bond, angle and dihedral terms plus non-bonded LJ, and Coulomb interactions that have been accurately determined for hydrocarbons (that is, ethanol). The thermodynamic and structural results from this model were shown to be in good agreement with available experimental and theoretical studies[39,40]. Reactive empirical bond order potential (AIREBO)[41] was used for the graphene sheets and the trapped hexadecane.

For trapped NaCl nanocrystal simulations, we used hybrid potentials including AIREBO[41] for graphene layers, EIM[42] potential for NaCl crystal, and 12-6 LJ between ions and carbon atoms (as they are implemented in large-scale atomic/molecular massively parallel simulator[37]). For carbon or hydrocarbon systems in which chemical reactions are of interest, and which may also require non-bonded interactions, the AIREBO many-body potential provides an effective and accurate method for our molecular simulations[41]. The used force field provides both carbon–carbon stretching and bending energy terms as well as bond dissociation/formation. On the other hand, the EIM potential captures also charge-transfer effects and environment dependence of the ionic bonding[42]. Regarding the multi-body nature of EIM, it provides a more realistic description of the ionic compounds than the more common models, which simply use Coulomb and vdW interactions between ions with fixed point charges. The cross LJ potential parameters were obtained by the Lorentz–Berthelot combining rules ($\varepsilon_{Na-C} = 0.006505$ kcal mol$^{-1}$, $\sigma_{Na-C} = 2.78$ Å, $\varepsilon_{Cl-C} = 0.001153$ kcal mol$^{-1}$, $\sigma_{Cl-C} = 4.115$ Å)[31]. The cutoff potential for the LJ potential was chosen at 10 Å.

Our simulation setup comprises three different parts: (i) the square shape substrate which is a rigid graphene sheet with 120,000 carbon atoms, (ii) a circular flake of graphene with 72,000 atoms with a typical radius of 17 nm and (iii) the trapped molecules, see Supplementary Fig. 8. In order to study the deformation of the top graphene layer, we first deformed it manually so that it covers the molecules underneath, then we performed an annealing MDs simulation by cooling down the system to 0 K. The latter optimizes the bump and the trapped molecules and allows us to achieve the minimum energy configuration. In the second step, we heat the system until room temperature. The Nosé–Hoover thermostat is used with time step 0.5 fs. The boundary around the circular flake is terminated by hydrogen in order to keep them chemically inactive. Note that the radius of the flake is taken large enough as compared to the bump radius ($4\times$) in order to avoid edge effects. We set the number of atoms for each bubble type such that we obtain approximately the same bubble size. This allows us to compare the different bubble types with each other. Visual molecular dynamics (VMD) package has been used to visualize the atoms/molecules[43].

**AFM measurements.** Graphene nanobubbles filled with water/ethanol molecules were fabricated by wet transfer technique using single and few layer graphene flakes (50 μm × 50 μm or above) prepared on the oxidized silicon substrate via mechanical exfoliation, as described previously[8]. In brief, single-layer graphene supported on poly (methyl methacrylate) (PMMA) layer prepared by wet etching method, was used as a top layer to enclose the solvent (water/ethanol) placed on another few layer graphene or graphite flake prepared on SiO$_2$/Si substrate. A well-controlled micro-manipulation setup was used to transfer the top layer for successful encapsulation of the solvent. After placing the single-layer graphene on top of 2 μl solvent, most of it was spontaneously squeezed out by leaving only a very small amount in between the top and bottom graphene layers. Overnight drying of the prepared samples at room-temperature led to the gradual evaporation of the solvent which allows the top graphene layer to completely collapse onto the bottom graphene flake with a tiny amount of solvent captured in between. These samples were placed in vacuum ($\sim 1$ mbar) for few hours before removing the PMMA layer using acetone wash. As prepared sandwich samples, containing the water/ethanol filled graphene nanobubbles, were used for atomic force microscope imaging using Bruker Dimension Fastscan AFM operating in peak force tapping mode. In-plane radius of the bubble in this study refers to the base radius measured using AFM profile. Height profile is taken across the bubbles by allowing the measuring line to pass through the centre of the bubble.

**Data availability.** The data that support the findings of this study are available from the corresponding author on request.

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

## Acknowledgements

We acknowledge fruitful discussion with Irina Grigorieva and Andre K. Geim. This work was supported by the Flemish Science Foundation (FWO-Vl) and the Methusalem program, the Royal Society and the Engineering and Physical Sciences Research Council, UK (EP/K016946/1). M.N.-A. was supported by Iran National Science Foundation (INSF).

## Author contributions

This project was designed by M. N.-A and directed by F.M.P. and M.N.-A. H.G.-K. carried out molecular dynamics simulations. R.R.N. and K.S.V. prepared the samples and performed the AFM and Raman measurements and data analysis. All authors contributed to discussions and the writing of the manuscript.

## Additional information

**Competing interests:** The authors declare no competing financial interests.

