## [Peer Review File · Nature Communications]

Reviewers' comments:

Reviewer #1 (Remarks to the Author):

The authors try to predict the shape of graphene nanobubbles using molecular dynamics simulations of trapped molecules within graphite flakes. Shapes are compared with AFM measurements, either performed by the authors or by others. One of the key innovative results of the method, in my opinion, is the possibility of determining the content of graphene bubbles by their shape (either in terms of the potential chemical and of its state). The authors use this to discuss their results vs. current state of the art.

The study is well written and excellently put in context: as such it should be considered for publication. Before being accepted by Nature Communications, however, I believe that some key points need to be addressed better. In particular:

[General] I could not spot any significant difference between the manuscript and the "relevant" publication...but equally I could not find any description of where the "relevant" publication has appeared. Please clarify.

[General] The authors spend very few words on the potential impact of their study, while this should be a crucial aspect for journals of the Nature family. In particular, how the authors see future studies to be changed by their findings? What is the effect of "bumps" (as they are mentioned in the text) on manufactured graphene flakes? Does their shape matter? If bump shape is determined by the trapped molecules, can we influence it by changing manufacturing conditions?

[General] Would it be possible to report how the obtained densities compare with the bulk values at 300 K? Such comparison seems to be there only for He.

[General] Molecular modelling consistently predicts sizes which are smaller than the experimental ones reported in this study. While I agree that a thorough explanation may be out of the scope for this study, this is a crucial aspect in comparing with other studies. Would the authors please provide an explanation? With (much) larger bubbles the shape is determined by a balance between the external pressure, the shell elastic properties and the internal pressure. A larger bubble seems to imply that the internal pressure has been underestimated by the current model. Is this a limit of the model?

[General] Bubbles due to ethanol and water are described as "spherical". This however refers only to the XY symmetry: the z height is consistently smaller than the radius. Would "Cylindrical symmetrical" be better? Otherwise, please explain why.

[General] Please state clearly in the figures which are models and which are experiments.

[General] There is a hot discussion at the moment in the metrological community on how to define "the radius of a bubble". Please state how it is defined here (i.e. it seems to be the end of the predicted/measured AFM profile)

[Introduction] The first paragraph of the introduction states that determining the pressure is still a challenging problem...so the reader expects a solution proposed by this study. When eq. (1) and (2) are presented, the reader is led to believe that eq. (2) is better for smaller bubbles...while at page 16 it is said that we can use e. (1). Please adjust text accordingly, either in the introduction or later.

[Page 5, line 10] Reference missing before eq. (3)

[Page 5, eq. (4)] I suggest putting the Trace operator in normal text mode, otherwise it is easy to read $T \cdot r(\sigma)$

[Page 6, line 11] "occur in the top layer": should it be "occur between"?

[Page 6, line 13] The He bubble is said to evolve in a non-spherical shape, but the XY profile are identical. Please clarify the text refers to z in this case.

[Page 8, part on He]. The obtained density is said to be consistent with the one of liquid Helium. However, the RDF profile is said to have only one peak. With ethanol, later, the fact that ethanol stays liquid is justified by the appearance of a second peak. Is He different? Or does Figure 2 contain two peaks instead at 300 K?

[Page 8, last line] Is "size" equal to "radius" or something else?

[Page 9, last line] please cite explicitly the size of the bubbles and/or the conditions used in ref

[29] to justify the comparison.

[Page 11, Figure 5] How do these results compare with the layered case discussed in ref [30]?

[Page 11] Please insert the expected density for ethanol.

[Page 2, paragraph starting at line 3] Please cite the manufacturing method described in the methods section, otherwise it is not clear how you are sure of what is in your bubbles.

[Page 14, 6 lines from bottom] change "reported" into "reported in Table I"

[Page 14, last line] The authors conclude that a universal scaling law is not valid...but this is not summarised in the conclusions.

[Page 15, figure 9] I find this figure confusing...as it is not possible to determine the height of the bumps from the scale on the left. Would it be possible e.g. to report the profiles on the same vertical scale?

[Page 16, eq. (6)] Please clarify what is x.

[Page 16, Table II] I could not find a definition of Q.

[Page 17, line 7, "optimized structures"] Should it be "shapes" here?

[Page 17, line 9] It is stated here that the fact that shape does not evolve with temperature is unexpected. How does this impact on the conclusions? Is this a limit of the model or something else?

[Page 17, line 12, "we found that by increasing..."] Please clarify how this sentence impacts on the rest of the paragraph. Does it mean that we should not worry about the fact that e.g. triangular shapes stay the same while passing from 0 K to 300 K, as the bubbles considered in this study are small?

[Page 17, 4 lines from end, "trivial, but our calculations, for the first time, reveal the atomistic details for this phenomenon"] Please remove "but" and "for the first time". Change "phenomenanan" with either "phenomenon" or "phenomena".

[Page 18, Figure 11] It would be crucial to have here an experimental comparison. In any case, please clarify these are modelling predictions.

[Page 19, end of section], substitute "ic" with "is"

[Page 19, section on adhesion energy] The impact of these result on the general discussion is not sufficiently clear. They seem to anticipate a potential comment on the method, but the relevance of Figures 12 and Figure 13 is undersold.

[Page 21, start of Conclusions] Please remove "Motivation by" and change "has" into "have".

[Page 22, "the water nanobubble has a circular shape"] So, is it spherical as said here or flattened as mentioned in the H2O section?

[Page 23] A different number of molecules has been chosen for the different bubble contents. Is there a methodology behind or is this somewhat arbitrary? Does the result change if e.g. one could use more molecules?

[Page 25] Please clarify which solvent was used.

[Supplementary] The text mentions supplementary videos, but since they don't seem to be part of this submission, it should be clearly said where they are.

Reviewer #2 (Remarks to the Author):

The paper 'Graphene nanobubbles of Helium, Water, Hydrocarbons and NaCl: shape, pressure and substance dependence' discussed the structures & pressure of nanoconfined helium, water, hydrocarbons and NaCl clusters based on a combined study using elastic theory, molecular simulations and AFM characterization. The general conclusions include that the confined phases demonstrate layered structures, although the molecular-level ordering is different in these substances. However, my general feeling is that the discussion of results on the theoretical, simulation and experimental data are not well integrated, and thus prohibit offering in-depth insights into the thermodynamics of condensed matter confined in the nanobubbles. There have been quite a number of theoretical and molecular simulation studies recently that explored the structures and thermodynamics of nanoconfined fluids/solids, and the results here do not provide enough new materials/insights, at least in the current form of presentation. Consequently, I cannot recommend the present manuscript for publication in Nature Communications, and here

are some detailed comments that may help the authors to improve their work.

(1) The predictions of pressure (Eqs. 1 and 2) are based on the membrane theory, which does not include the contribution/effect from elasticity of the nanoconfined matter. The predictions thus should be updated, especially in this work that discuss gas, liquid, solid phases of confined matter in the nanobubbles. Moreover, on Page 5, it was mentioned that the stress tensor is analyzed in this work. However, I did not find discussion on the results in the following content. The analysis of stress in the confined matter (based on the MD simulation results) not only provides data that can be used to validate theoretical predictions (Eqs. 1 and 2), but also helps to clarify the phases (e.g., gas and liquid do not resist shear stress, but solid does).

(2) The layered structures of confined helium, water and hydrocarbon should be elaborated. There are 1-3 peaks in the density profile plotted along the thickness direction, would the layered structures be stable if the volume of nanobubbles increase? What is the major driving force for this order, the nanoconfinement or the pressure?

(3) In the results and discussion, the results from elastic theory, molecular simulations and experimental characterization were introduced without specification. To my understanding, the pressure was predicted from theory, the RDFs were calculated from MD simulated atomic structures and the height profiles were measured by AFM. However, there is no comparative discussion on these three approaches – for example, how do the pressure and stress tensor predicted from theory compare with the MD simulated results? how could the height profiles predicted from theory and the simulations be compared to the AFM measurement? Could the theoretical understanding be used to extract structural and pressure information from the AFM measured data?

(4) The molecular structures of nanoconfined phases are not well analyzed and presented, either from molecular simulations or the experimental data.

Reviewer #3 (Remarks to the Author):

Referee report

Title: Graphene nanobubbles of Helium, water, hydrocarbons and NaCl: shape, pressure and substance dependence

By: Ghorbanfekr-Kalashami et al.

This is a well-written and scientifically sound manuscript on graphene nanobubbles. Triggered by experimental studies on the shape and physical properties of graphene nanobubbles the authors performed a thorough and combined experimental/theoretical (atomic force microscopy/molecular dynamic simulations) study. The authors have considered five different nanobubbles substances namely: water, helium ethanol, hexadecane and NaCl. Depending on the substance the authors found nanobubbles that are circular and flexible, i.e. gel-like, (helium), to rigid (NaCl) and flat (hexadecane) nanobubbles, spherical (water, ethanol).

Major remarks/questions

1) On page 2, line 3: why is there a minimum nanobubble height of 0.34 nm (monolayer step height)?

2) I would guess that also the substrate has a profound effect on the ordering of the molecules/atoms of the nanobubble substance. It is not clear for me why this effect does not play a role here. Can the authors comment on this point?

3) In their analysis the authors taken into the hydrostatic pressure and the adhesion energy of the graphene bubbles. I do not understand why the authors have not considered the stretching of the graphene. Stretching of the graphene energy costs energy and will contribute to the formation of the graphene nanobubbles. Can the authors comment on this?

Minor remarks (typos, grammatical errors etc.)

1) Page 2, line 3 (introduction)have size from .. -> have sizes from....

2) Page 2, line 5: on the mica substrates..... -> on mica substrates.

In summary, this is an interesting and relevant paper. After revision the paper might be suitable for publication. Although I'm not sure if this manuscript fully meets all the requirements for publication in Nature Communications I tend to give the paper the benefit of the doubt.

Reply to Reviewer # 1**General comments by the reviewer:**

The authors try to predict the shape of graphene nanobubbles using molecular dynamics simulations of trapped molecules within graphite flakes. Shapes are compared with AFM measurements, either performed by the authors or by others. One of the key innovative results of the method, in my opinion, is the possibility of determining the content of graphene bubbles by their shape (either in terms of the potential chemical and of its state). The authors use this to discuss their results vs. current state of the art. The study is well written and excellently put in context: as such it should be considered for publication. Before being accepted by Nature Communications, however, I believe that some key points need to be addressed better.

Our reply:

We would like to thank the referee for very careful reading our paper and the very useful comments.

Comments:**Reviewer # 1****[COMMENTS REGARDING ENUMERATING]**

[General] I could not spot any significant difference between the manuscript and the “relevant” publication...but equally I could not find any description of where the “relevant” publication has appeared. Please clarify.

Our reply:

We added the following text to the “Introduction”:

“Previous works concentrated on general geometrical aspects of graphene bubbles [10, 13] where the content of the bubble was gas or liquid-like. Here, for the first time, we focus on how the material inside the bubble influences the microscopic shape of the bubble. For illustrative purposes, we considered four materials with very different properties. Hydrocarbons are often present as contaminants [10, 13]. Several studies exist on confined water [16–19] while ethanol, helium and NaCl are rarely considered.”

Using state of the art molecular dynamics simulations, we elucidate the differences in the shape of deformed graphene for these different materials. The layered structure of these four materials are completely different. Helium has a layered structure only at very low temperature while water and ethanol preserve their layered structure even at room temperature. Graphene over NaCl always follows the crystal shape, except if it is solved in a solution. Bubbles filled with hydrocarbon follow very well elasticity theory in consistent to other used materials, and are therefore suitable test materials for studies based on elasticity theory and secondly, there are also various accurate force fields in MD simulations for hydrocarbons which makes it possible to simulate them accurately. The microscopic structure of trapped water and ethanol are very different which are due to the different distribution of H-bonds in ethanol.

Reviewer # 1

[General] The authors spend very few words on the potential impact of their study, while this should be a crucial aspect for journals of the Nature family. In particular, how the authors see future studies to be changed by their findings? What is the effect of “bumps” (as they are mentioned in the text) on manufactured graphene flakes? Does their shape matter? If bump shape is determined by the trapped molecules, can we influence it by changing manufacturing conditions?

Our reply:

We added the following text to the “Conclusions”:

“We presented an extensive atomistic molecular dynamics simulation study to probe the van der Waals (vdW) pressure at atomically smooth graphene interfaces by intentional trapping different types of materials (water, gas, hydrocarbons, and salt). The correlation between the shape of the graphene bubble and the physical properties of the trapped substances was pointed out. Our detailed simulations along with the experimental observations provides insightful information about the formation of bubbles (including flat nanobubbles) and the effect of vdW pressure on the structural and conformational changes of the trapped substance. By observing the shape of the bubble it should be possible to obtain information about physical and chemical changes that may occur in nano-enclosures under the influence of high pressure and different temperatures [33, 34].”

Reviewer # 1

[General] Would it be possible to report how the obtained densities compare with the bulk values at 300 K? Such comparison seems to be there only for He.

Our reply:

We now compared the obtained densities with those for bulk helium, water, and ethanol at room temperature at two different pressure 1 atm and the corresponding induced pressure for each bubble type (of the order of GPa). We added the below table into the text (TABLE II) in order to compare densities with those of bulk. The density of encapsulated substances are close to their bulk value when subjected to similar pressures. This is an additional supports for the fact that high vdW pressures are present inside the nanosize graphene bubble.

substance	P_{vdW} (GPa) bubble	ρ (gr/cm ³) bubble	ρ (gr/cm ³) bulk	ρ (gr/cm ³) bulk P=1 atm
Helium	0.65	0.225	0.283 (P=0.65 Gpa)	0.0043
Water	0.93	1.103	1.191 (P=0.93 Gpa)	1.044
Ethanol	0.49	0.987	1.051 (P=0.49GPa)	0.862

Caption: The density of encapsulated substances and those for bulk.

Reviewer # 1

[General] Molecular modelling consistently predicts sizes which are smaller than the experimental ones reported in this study. While I agree that a thorough explanation may be out of the scope for this study, this is a crucial aspect in comparing with other studies. Would the authors please provide an explanation? With (much) larger bubbles the shape is determined by a balance between the external pressure, the shell elastic properties and the internal pressure. A larger bubble seems to imply that the internal pressure has been underestimated by the current model. Is this a limit of the model?

Our reply:

We added a new section and a new figure (Fig. 19-20) about the size effect. We also briefly reviewed the prediction from continuum theory; see added supplementary “Bubble elasticity”.

We added the following text at end of supplementary section “Molecular dynamics simulations”.

“We admit that without corrections for the effective volume of the bump due to difficulty in estimating the exact accessible volume for the trapped atoms, MD overestimates the measured volume. Consequently, it underestimates the pressure and the density. This effect becomes important for small bubbles when the size of the excluded volume is comparable to the total volume.”

Reviewer # 1

[General] Bubbles due to ethanol and water are described as “spherical”. This however refers only to the XY symmetry: the z height is consistently smaller than the radius. Would Cylindrical symmetrical be better? Otherwise, please explain why.

Our reply:

We changed “spherical” to “round shape” with cylindrical symmetry in XY-plane” everywhere in the text.

Reviewer # 1

[General] Please state clearly in the figures which are models and which are experiments.

Our reply:

We revised caption, figures, and corresponding text of figures 4 and 7 to clarify the experimental AFM and theoretical MD figures. We added two new experimental AFM images to the supplementary information.

Reviewer # 1

[General] There is a hot discussion at the moment in the methological community on how to define “the radius of a bubble”. Please state how it is defined here (i.e. it seems to be the end of the predicted/measured AFM profile)

Our reply:

For the experiment part, we added the following lines to the supplementary “Atomic force microscopy measurements”: “Radius of the bubble in this study refers to the “base radius”

measured using the AFM profile. Height profile is taken across the bubbles by allowing the measuring line to pass through the center of the bubble.”

The following text added to the supplementary.

Supplementary information “Bubble volume and radius calculations”

The pressure and density of the materials inside the bubble crucially depend on how accurately we determine the bump volume. Pressure and density inside the bubble can be found by calculating the bump volume using Eqs. (4,5). An accurate determination of the height and radius of the bubble is important in order to obtain the morphology of the bubble.

To calculate the volume of the bubble, the top graphene layer is fitted by a function $h(x,y)$ to determine the time average xyz coordinate of the carbon atoms. Because not the whole space of the volume is accessible to the trapped atoms -due to repulsive potential between graphene and the atoms- we defined the effective height function $h_{eff}(x,y)$ which excludes the volume related to the distance between the molecule-graphene (σ_{mol-c}):

$$h_{eff}(x,y) = h(x,y) - \sigma_{mol-c}$$

where σ_{mol-c} is average LJ parameter between trapped molecules and C atoms. Then, we estimated the volume and the radius by integrating over the bump area using the following equations:

$$V = \int_{bump} h_{eff}(x,y) dx dy$$

$$A = \int_{bump} dx dy \Rightarrow r = \sqrt{\frac{A}{\pi}}$$

The bump area is defined by the estimated region where the trapped atoms are located.

Reviewer # 1

1-[Introduction] The first paragraph of the introduction states that determining the pressure is still a challenging problem...so the reader expects a solution proposed by this study. When eq. (1) and (2) are presented, the reader is led to believe that eq. (2) is better for smaller bubbles...while at page 16 it is said that we can use e. (1). Please adjust text accordingly, either in the introduction or later.

Our reply:

We modified the text and clarified these statements. We extend the elasticity theory section by adding supplementary “Bubble elasticity”. In the revised version, we have a new prediction from elasticity theory.

Eqs. (1) and (2) result in barely noticeable different predictions due to the small contribution of the bending rigidity to the elastic energy of graphene beyond the length scale of 4\AA [13].

We added the following line after Eq. (2) to clarify this issue:

We emphasized in the introduction that our method based on the virial stress tensor is more general than Eqs. (1,2). The latter originates from elasticity theory with the assumption of a round-shape and using the large bubble limit. Accordingly, it in general fails to describe various bubble shapes such as

semi-circular or noncircular in our simulated system.

This is the reason which explains why the presented elastic models (membrane and nonlinear plate models) are in good agreement with our result in case of round shape ethanol bubble.

To clarify these issues we added new supplementary “Bubble elasticity”

Reviewer # 1

2-[Page 5, line 10] Reference missing before eq. (3)

Our reply:

Corrected, thank you.

Reviewer # 1

3-[Page 5, eq. (4)] I suggest putting the Trace operator in normal text mode, otherwise it is easy to read $T \cdot r(\sigma)$

Our reply:

Corrected.

Reviewer # 1

4-[Page 6, line 11] “occur in the top layer”: should it be “occur between”?

Our reply:

Corrected.

Reviewer # 1

5-[Page 6, line 13] The He bubble is said to evolve in a non-spherical shape, but the XY profile are identical. Please clarify the text refers to z in this case.

Our reply:

We modified the bump 2D mesh plot and corresponding line profile. These figures exhibit a single snapshot at particular time of the bubble. Generally, based on our MD simulations, we concluded that He forms a non-circular bubble. We modified figures 1(a,b,c).

Reviewer # 1

6-[Page 8, part on He]. The obtained density is said to be consistent with the one of liquid Helium. However, the RDF profile is said to have only one peak. With ethanol, later, the fact that ethanol stays liquid is justified by the appearance of a second peak. Is He different? Or does Figure 2 contain two peaks instead at 300 K?

Our reply:

Helium under GPa pressure cannot be in the gas phase. On the other hand, the solid phase characterized by long range order. Here, we found that helium has two peaks where the second one

is small. Therefore we are reluctant to state that it is in the liquid phase but for sure it is not in the gas phase. The formation of a layer close to the substrate motivated us to say that a mixture of liquid and solid phases is present/or a phase separation. At the top, it has less density, and it is more close to the gas phase.

In order to exaggerate the second peak of trapped helium, instead of using the usual RDF, we calculated the lateral radial distribution function [Phys. Rev. E 72, 051503 (2005)] along the direction perpendicular to the substrate (z-axis). RDF and lateral-RDF are identical for bulk system. However, for our encapsulated system, the latter corrects the effect of confinement in the z-direction and this is the reason why we now see a clear second peak in RDF figure 3(c).

Lateral RDF calculation is calculated as follows: Assuming r_{ij} are xy-plane radial distance and using $\delta z = 0.1$ nm, we defined

$$g_{||}(r) = \text{coef.} \sum_{i \neq j} \delta(r - r_{ij}) [\theta(z_i - z_j + \delta z/2) - \theta(z_i - z_j - \delta z/2)]$$

The corresponding figures (1(c), 3(c)) were modified.

Reviewer # 1

7-[Page 8, last line] Is "size" equal to "radius" or something else?

Our reply:

It is size. We rephrased it.

Reviewer # 1

8-[Page 9, last line] please cite explicitly the size of the bubbles and/or the conditions used in ref [29] to justify the comparison.

Our reply:

We added some explanation related to Ref. 19 and compared our results with this experiment. We change the text as follows:

"In contrast to Ref. [24], we argue that the density and size of water bubbles are important factors that control the microscopic structure of trapped water which was not considered in other theoretical work [16]. In the latter study, confined water between two parallel graphene sheets was studied for different interlayer distances (6-12 Å) using MD simulations. At high pressure (~ 1 GPa), they observed AA stacking of two layers of square ice while in our simulated bubble we found AB stacking as in Ref. [17]."

Reviewer # 1

9-[Page 11, Figure 5] How do these results compare with the layered case discussed in ref [30]?

Our reply:

We compared and explained our results and pointed out the differences between the two works of the end of Sec. III-F. See also new supplementary “Layered structure”.

Reviewer # 1

10-[Page 11] Please insert the expected density for ethanol.

Our reply:

We compared the density of trapped ethanol and its bulk for two different pressures and listed results in new Table II. The expected density is comparable with bulk density at high pressure (order of GPa). At low pressure, the density of encapsulated ethanol is larger than that for bulk at normal condition.

Reviewer # 1

11-[Page 2, paragraph starting at line 3] Please cite the manufacturing method described in the methods section, otherwise it is not clear how you are sure of what is in your bubbles.

Our reply:**Supplementary information “Solvents” and two corresponding figures.**

“We have used water, ethanol and methanol as solvents to fabricate solvent filled graphene nanobubbles. We presented the data and AFM images of water/ethanol filled with graphene nanobubbles in Figs. 4(a,b) and 7(a,b) within the main text. We have also carried out Raman spectroscopy as a characterization technique to probe the presence of trapped substances. As a reference, here we have shown the Raman spectrum of bulk methanol/ethanol and methanol/ethanol filled graphene nanobubbles (see Fig. 24). It is clear that all the Raman bands related to the methanol/ethanol (indicated as red colour arrow marks in the inset figures) are present in the corresponding nanobubbles with small shifts in the wavenumbers.”

FIG. 24: (Color online) Raman spectrum of bulk methanol/ethanol and methanol/ethanol filled graphene nanobubbles (Inset figures).

Reviewer # 1

12-[Page 14, 6 lines from bottom] change “reported” into “reported in Table I”

Our reply:

Corrected.

Reviewer # 1

13-[Page 14, last line] The authors conclude that a universal scaling law is not valid...but this is not summarised in the conclusions.

Our reply:

We added the following lines to the conclusion before the acknowledgment.

“Finally, we concluded that universal scaling law is limited to round shape bubbles such as found for a graphene bubble filled with hydrocarbons. But, generally such scaling law is not applicable for any bubble type. As an extreme case, NaCl bubble does not follow the universal scaling law mentioned in Ref. [13]. Our study provides fundamental insights into the formation of graphene nanobubbles and the effect of microscopic details of the trapped substances on it.”

Reviewer # 1

14-[Page 15, figure 9] I find this figure confusing...as it is not possible to determine the height of the bumps from the scale on the left. Would it be possible e.g. to report the profiles on the same vertical scale?

Our reply:

We updated that figure with separate vertical scales (see the following figure).

Reviewer # 1

15-[Page 16, eq. (6)] Please clarify what is x.

Our reply:

We defined x as a scaled radial distance ($x=r/R$), where r (R) are radial distance (radius) of the bubble.

Reviewer # 1

16-[Page 16, Table II] I could not find a definition of Q.

Our reply:

We defined Q in the text after Eq. (7).

Q as quality factory is defined by, where $Q = 1 - \sqrt{\frac{\sum_{i=1}^N (\sigma_i - f(x_i))^2}{N}}$

Similar definition used for Table IV in the supplementary: $Q = 1 - \sqrt{\frac{\sum_{i=1}^N (h/h_{\max} - f(x_i))^2}{N}}$

Reviewer # 1

17-[Page 17, line 7, “optimized structures”] Should it be “shapes” here?

Our reply:

Yes, shape is also a nice word, we used it and explain what is the optimization.

Reviewer # 1

18-[Page 17, line 9] It is stated here that the fact that shape does not evolve with temperature is unexpected. How does this impact on the conclusions? Is this a limit of the model or something else?

Our reply:

Though the question is not completely clear for us but we argue that, often we average out the deformation in time we find almost the same structure.

Reviewer # 1

19-[Page 17, line 12, “we found that by increasing...”] Please clarify how this sentence impacts on the rest of the paragraph. Does it mean that we should not worry about the fact that e.g. triangular shapes stay the same while passing from 0 K to 300 K, as the bubbles considered in this study are small?

Our reply:

We emphasized the fact that the initial shape of NaCl bubble does not change when increasing the size even when T is up to room temperature (see below Fig. 11).

Reviewer # 1

20-[Page 17, 4 lines from end, “trivial, but our calculations, for the first time, reveal the atomistic details for this phenomenon”] Please remove “but” and “for the first time”. Change

“phenomenanan” with either “phenomenon” or “phenomena”.

Our reply:

Corrected.

Reviewer # 1

21-[Page 18, Figure 11] It would be crucial to have here an experimental comparison. I any case, please clarify these are modelling predictions.

Our reply:

We modified text and corresponding caption of Fig. 11 to emphasize that these are simulation results.

New caption for Fig. 11

“MD predicted side (a) and top (b) views of the minimized energy of encapsulated square NaCl at 0 K. (c) The corresponding partial Na-Na and Na-Cl radial distribution function. The simulated graphene flake and the graphene substrate are not shown here

Reviewer # 1

22-[Page 19, end of section], substitute “ic” with “is”

Our reply:

Corrected.

Reviewer # 1

23-[Page 19, section on adhesion energy] The impact of these result on the general discussion is not sufficiently clear. They seem to anticipate a potential comment on the method, but the relevance of Figures 12 and Figure 13 is undersold.

Our reply:

We added below line to clarify the impact of the results on the general discussion (see end of section “The effects of adhesion energy” in the supplementary).

“We have shown that even by increasing the adhesion energy between the two graphene sheets, which is equivalent to increasing the vdW pressure, a more ordered structure for water is obtained, however the square ice structure and its stacking structure are not seen. This casts some doubts on the observation of square ice in Ref. [29].”

Reviewer # 1

24-[Page 21, start of Conclusions] Please remove “Motivation by” and change “has” into “have”.

Our reply:

Corrected.

Reviewer # 1

25-[Page 22, “the water nanobubble has a circular shape”] So, is it spherical as said here or flattened as mentioned in the H2O section?

Our reply:

Water bubble is semi-circular and not flattened. We correct it in the text.

Reviewer # 1

26-[Page 23] A different number of molecules has been chosen for the different bubble contents. Is there a methodology behind or is this somewhat arbitrary? Does the result change if e.g. one could use more molecules?

Our reply:

We added few lines to clarify the different choices of molecules inside bubbles as follows.

“We set the number of atoms for each bubble type such that we obtain approximately the same bubble size. This allows us to compare the different bubble types with each other. As the number of molecules increased, we expect larger bubble, more number of layers, and smaller pressure. But, based on our MD simulation, we do not expect fundamental changes in bubble shape for nanosize range of bubbles such as e.g. RDF and scaled morphology.”

Reviewer # 1

27-[Page 25] Please clarify which solvent was used.

Our reply:

We have used water, ethanol and methanol as solvents to fabricate solvent filled graphene nanobubbles. We presented the data and AFM images of water/ethanol filled with graphene nanobubbles. See new supplementary “Solvent”.

Reviewer # 1

28-[Supplementary] The text mentions supplementary videos, but since they don't seem to be part of this submission, it should be clearly said where they are.

Our reply:

Sorry. We added them to the resubmitted files.

Reply to Reviewer # 2

General comments by the reviewer:

The general conclusions include that the confined phases demonstrate layered structures, although the molecular-level ordering is different in these substances. However, my general feeling is that the discussion of results on the theoretical, simulation and experimental data are not well integrated, and thus prohibit offering in-depth insights into the thermodynamics of condensed matter confined in the nanobubbles. There have been quite a number of theoretical and molecular simulation studies recently that explored the structures and thermodynamics of nanoconfined fluids/solids, and the results here do not provide enough new materials/insights, at least in the current form of presentation. Consequently, I cannot recommend the present manuscript for publication in Nature Communications, and here are some detailed comments that may help the authors to improve their work.

Our reply:

We would like to thank the referee for careful reading of our paper and the very useful comments. We substantially revised our paper and added many new results (experimental and theoretical). We also provide a new supplementary containing many new sections.

Comments:

Reviewer # 2

1-The predictions of pressure (Eqs. 1 and 2) are based on the membrane theory, which does not include the contribution/effect from elasticity of the nanoconfined matter. The predictions thus should be updated, especially in this work that discuss gas, liquid, solid phases of confined matter in the nanobubbles. Moreover, on Page 5, it was mentioned that the stress tensor is analyzed in this work. However, I did not find discussion on the results in the following content. The analysis of stress in the confined matter (based on the MD simulation results) not only provides data that can be used to validate theoretical predictions (Eqs. 1 and 2), but also helps to clarify the phases (e.g., gas and liquid do not resist shear stress, but solid does).

Our reply:

First, we added new supplementary "Bubble elasticity" about elasticity theory background of Eqs. (1,2). We also added three new subsections to the manuscript which contain many new results about stress tensor calculations and corresponding properties, i.e. stress, size effects, and layered structures. We calculated the diagonal element of the stress tensor for the studied bubbles. We compared our results with those predicted by elasticity theory and discuss the results.

Reviewer # 2

2-The layered structures of confined helium, water and hydrocarbon should be elaborated. There are 1-3 peaks in the density profile plotted along the thickness direction, would the layered structures be stable if the volume of nanobubbles increase? What is the major driving force for this order, the nanoconfinement or the pressure?

Our reply:

We added a new section about the structure of trapped materials inside the bubbles and discuss the results in section “Layered structure”.

We also compared our results with those from other works (see added sections). When the volume of the bubbles increases the layered structure is found near the boundaries. The major force behind it is the confinement. Notice that the large pressure is also a consequence of the confinement.

Reviewer # 2

3-In the results and discussion, the results from elastic theory, molecular simulations and experimental characterization were introduced without specification. To my understanding, the pressure was predicted from theory, the RDFs were calculated from MD simulated atomic structures and the height profiles were measured by AFM. However, there is no comparative discussion on these three approaches – for example, how do the pressure and stress tensor predicted from theory compare with the MD simulated results? how could the height profiles predicted from theory and the simulations be compared to the AFM measurement? Could the theoretical understanding be used to extract structural and pressure information from the AFM measured data?

Our reply:

We added a few lines for comparing/discussing experimental and molecular simulation results. See newly added sections. The structural information from MD simulations predicts the AFM measured data for the structure. AFM images do not give us any information about pressure inside the bubbles.

Reviewer # 2

4-The molecular structures of nanoconfined phases are not well analyzed and presented, either from molecular simulations or the experimental data.

Our reply:

We added a new section about structure of confined water and ethanol, see “Layered structure” section.

Reply to Reviewer # 3

General comments by the reviewer:

This is a well-written and scientifically sound manuscript on graphene nanobubbles. Triggered by experimental studies on the shape and physical properties of graphene nanobubbles the authors performed a thorough and combined experimental/theoretical (atomic force microscopy/molecular dynamic simulations) study. The authors have considered five different nanobubbles substances namely: water, helium ethanol, hexadecane and NaCl. Depending on the substance the authors found nanobubbles that are circular and flexible, i.e. gel-like, (helium), to rigid (NaCl) and flat (hexadecane) nanobubbles, spherical (water, ethanol).

Our reply:

We would like to thank the referee for very careful reading of our paper and the very useful comments.

Major comments:

Reviewer # 3

1-On page 2, line 3: why is there a minimum nanobubble height of 0.34 nm (monolayer step height)?

Our reply:

We added this to line 3 page 2:

“This is the minimum observed height of a monolayer of atomically flat water adlayer on the mica substrate.”

Reviewer # 3

2-I would guess that also the substrate has a profound effect on the ordering of the molecules/atoms of the nanobubble substance. It is not clear for me why this effect does not play a role here. Can the authors comment on this point?

Our reply:

We added a new section about layered structure and explained this issue.

Reviewer # 3

3-In their analysis the authors taken into the hydrostatic pressure and the adhesion energy of the graphene bubbles. I do not understand why the authors have not considered the stretching of the graphene. Stretching of the graphene energy costs energy and will contribute to the formation of the graphene nanobubbles. Can the authors comment on this?

Our reply:

In our MD simulations, we used classical force fields and therefore it automatically includes both stretching and bending energy by employing various terms in the force field. In the method section we expanded on this issue.

Minor comments:**Reviewer # 3**

1-Page 2, line 3 (introduction)have size from .. -> have sizes from....

Our reply:

Corrected.

Reviewer # 3

2-Page 2, line 5: on the mica substrates..... -> on mica substrates.

Our reply:

Corrected.

REVIEWERS' COMMENTS:

Reviewer #1 (Remarks to the Author):

I find this revised version of their study, where the authors have fully addressed most of the reviewers' comments, surely much stronger than the previous. Thanks. I recommend, however, that some additional (minor) corrections are made before publication. Here are some comments in this direction.

[Introduction, end of first paragraph] I would suggest to substitute "controlled" with "observed" or "monitored"

[Introduction, second paragraph] possible typo: it should be "a universal scaling" instead than "on universal scaling"

[Introduction, "recently, Khestanova et al."] Since part of the novelty of this work consists in the use of different trapped substances and considering that the trapped substance and the substrate clearly affect the experimental results, I believe it is important to state what was used in previous studies. Therefore, please specify which substances were used in Ref. [13] and in which case was found that bubbles below 50 nm did not follow membrane theory. Finally, the title reported in ref. [13] is not the one of Khestakova's Nat Comms paper.

[Introduction, "Following a similar theoretical approach..." and "For the smaller bubbles this universal"] These sentences now seem redundant, as the study by Khestanova et al. is detailed already one sentence earlier. Otherwise, maybe the authors are thinking of citing an additional reference where the law in Ref. [10] and [13] has been proven to be wrong below 50nm?

[Introduction, "Moreover, for a nanobubble"] As before, please specify which trapped substance was used in Ref. [14]. Also, consider rephrasing the sentence to make it clearer what causes the large pseudo-magnetic field.

[Introduction, end of second paragraph"] Potential typo: check "has been remained"

[Introduction, 4th paragraph, "NaCl are rarely considered"] Have results in He, ethanol and NaCl ever been presented? In positive case, please cite appropriate references.

[Page 3, start of section III.A] Please change "put" with "simulate"

[Page 3, Figure 1] In the rebuttal letter as in the text, the authors clarify that this is a snapshot in time of Helium evolution...and that at the end of the evolution the bubble is no longer symmetric. May be worth doing this in the caption too. This said, since the reader is not given any picture of an evolved bubble, the helium bubble looks nicely cylindrical symmetric...the fact that this symmetry is lost is not justified by (presented) data. Would it be possible e.g. to present a picture on the simulated equilibrium shape of a He cloud?

[Page 5, caption of figure 4, "results for large water bubble"] the use of "large" is not consistent with the introduction, where "large" = ">50 nm". Please clarify.

[Page 6, first paragraph] Please specify the conditions in study [29] for a fairer comparison.

[Page 6, second paragraph, "qualitatively in agreement"] Please clarify what is intended here with qualitative agreement e.g. cite more clearly the results in [30]

[Page 6, last sentence] please clarify whether this summarising sentence refers to simulations, measurements or both.

[Page 7, first paragraph] The authors say that the different shape is due to the higher viscosity in ethanol. This parameter does not appear in the membrane theory, so it may be worth clarifying (e.g. in the supplementary) where viscosity comes into play and what is the meaning of viscosity as such low temperatures. This is a crucial point because, as later stated at page 8, "only for ethanol elasticity theory provides nice fits to the MD data".

[Page 8, Table II] This table is different from the one in the rebuttal letter. Please check.

[Page 9, new figure 9] The new figure now clarifies that the sentence "hexadecane is flatter" does not mean that h_{\max}/R is lower, but that "there is a flat region on top of the hexadecane bubble". Please adjust the text accordingly, if possible.

[Page 9, last sentence] I would suggest removing "Notice".

[Page 11, section F, line 3] possible typo: should be "obtained". Also, this sentence is not clear.

[Page 11, section F, sentence starting with "Two RDFs are nicely fit"] Please check this sentence and remove "nice", which may sound too colloquial.

[Figures 13, 16, 18] These make the layered results much clearer! If possible, however, consider that they may be printed in B/W.

[Conclusions, general] Conclusions now start with impact, and this is good. After that, however, the reader is presented with a list of claims and the bigger picture (i.e. "we run extensive MD that allowed us to go beyond membrane theory, covering substances studied only by a few") may get lost. It may be worth spending a little more effort to transform the conclusions in a sequential story.

[Conclusions, reference [35]] I suggest substituting "our" with "a" here

[Conclusions, paragraph starting with "the deformed graphene"] I suggest to remove "always" here, as it clashes with "except".

[Conclusions, last paragraph] remove "the important messages" and "secondly" and rephrase. Also, I am not sure that membrane theory can predict the flattening on hexadecane nanobubbles...and so I would be careful in saying that these are the best candidate for studying elasticity theory. Please check.

[Conclusions, last paragraph] In my understanding, the sentence "such scaling law is not applicable to any bubble" is more correct than inferring it does not work for bubbles below 50nm, in general.

Reviewer #2 (Remarks to the Author):

Most of my previous concerns were addressed. However, I still feel that the conclusion from this work is not general and novel enough for Nature Communications – although I leave the judgement on these two factors to the editorial office.

The writing should also be improved. The presentation should be more concise, the conclusion should be sharpened, and the number of figures should be reduced in the main text.

Some additional comments:

1. 'D. Stress calculations',

why use J_2 as the indication of stress?

2. If the NaCl crystal is nucleated from solvent, would the triangular/ellipsoid/square NaCl be the favorable configurations under graphene, or the crystal will adapt the shape of graphene by

forming rough surfaces?

Reviewer #3 (Remarks to the Author):

The authors have satisfactorily addressed all my points/remarks and therefore I recommend to accept the manuscript in its present form.

Reply to Reviewer # 1

General comments by the reviewer:

I find this revised version of their study, where the authors have fully addressed most of the reviewers' comments, surely much stronger than the previous. Thanks. I recommend, however, that some additional (minor) corrections are made before publication.

Our reply:

We would like to thank the referee for the very careful reading of our paper and the very useful comments.

Comments:

Reviewer # 1

1. [Introduction, end of first paragraph] I would suggest to substitute "controlled" with "observed" or "monitored".

Our reply:

It was corrected into 'monitored'.

Reviewer # 1

2. [Introduction, second paragraph] possible typo: it should be "a universal scaling" instead than "on universal scaling".

Our reply:

It was corrected.

Reviewer # 1

3. [Introduction, "recently, Khestanova et al."] Since part of the novelty of this work consists in the use of different trapped substances and considering that the trapped substance and the substrate clearly affect the experimental results, I believe it is important to state what was used in previous studies. Therefore, please specify which substances were used in Ref. [13] and in which case was found that bubbles below 50 nm did not follow membrane theory. Finally, the title reported in ref. [13] is not the one of Khestakova's Nat Comms paper.

Our reply:

It was mentioned that the bubbles were filled with hydrocarbons which we now specify in the text.

The title of Ref 13 was corrected in the list of references.

Reviewer # 1

4. [Introduction, "Following a similar theoretical approach..." and "For the smaller bubbles this universal"] these sentences now seem redundant, as the study by Khestanova et al. is detailed already one sentence earlier. Otherwise, maybe the authors are thinking of citing an additional

reference where the law in Ref. [10] and [13] has been proven to be wrong below 50nm?

Our reply:

The mentioned sentences were removed.

Reviewer # 1

5. [Introduction, “Moreover, for a nanobubble”] as before, please specify which trapped substance was used in Ref. [14]. Also, consider rephrasing the sentence to make it clearer what causes the large pseudo-magnetic field.

Our reply:

We modified as the following:

“Moreover, for a nanobubble filled with ethylene with radius $R=4\text{nm}$ and height 0.5nm it was shown that this results in a pseudo magnetic field of 100 Tesla [14] which changes fundamentally the electronic spectrum of graphene. The latter is a consequence of the large induced strain in the graphene nanobubble.”

Reviewer # 1

6. [Introduction, end of second paragraph”] Potential typo: check “has been remained”.

Our reply:

It was corrected.

Reviewer # 1

7. [Introduction, 4th paragraph, “NaCl are rarely considered”] Have results in He, ethanol and NaCl ever been presented? In positive case, please cite appropriate references.

Our reply:

We did not find any references for bubbles filled with He, ethanol or NaCl and therefore we replaced ‘are rarely’ by ‘have not been’.

Reviewer # 1

8. [Page 3, start of section III.A] Please change “put” with “simulate”.

Our reply:

It was corrected

Reviewer # 1

9. [Page 3, Figure 1] In the rebuttal letter as in the text, the authors clarify that this is a snapshot in time of Helium evolution...and that at the end of the evolution the bubble is no longer symmetric. May be worth doing this in the caption too. This said, since the reader is not given any picture of an evolved bubble, the helium bubble looks nicely cylindrical symmetric...the fact that this symmetry is lost is not justified by (presented) data. Would it be possible e.g. to present a picture on the

simulated equilibrium shape of a He cloud?

Our reply:

Figures 1 (a,b) are updated following the comments from the referee. A snapshot of an asymmetric helium bubble is now given.

Reviewer # 1

10. [Page 5, caption of figure 4, “results for large water bubble”] the use of “large” is not consistent with the introduction, where “large” = “>50 nm”. Please clarify.

Our reply:

The word “large” was removed and the sentence was corrected.

Reviewer # 1

11. [Page 6, first paragraph] Please specify the conditions in study [29] for a fairer comparison.

Our reply:

We modified text as “It is noteworthy to look at the microscopic structure of trapped hydrocarbons. We found very good agreement between our MD results for hydrocarbon bubbles, the predictions from elasticity theory (Eqs. (1,2)).”

Reviewer # 1

12. [Page 6, second paragraph, “qualitatively in agreement”] Please clarify what is intended here with qualitative agreement e.g. cite more clearly the results in [30].

Our reply:

We added below line to cite more clearly the results in [30] :

“This is consistent with the fact that ethanol crystallizes for pressures around 1.5 GPa [30]. Using numbers given in Table I, Eq. (2) gives ”.

Reviewer # 1

13. [Page 6, last sentence] please clarify whether this summarizing sentence refers to simulations, measurements or both.

Our reply:

We added “In our molecular dynamics simulations ...”.

Reviewer # 1

14. [Page 7, first paragraph] The authors say that the different shape is due to the higher viscosity in ethanol. This parameter does not appear in the membrane theory, so it may be worth clarifying (e.g.

in the supplementary) where viscosity comes into play and what is the meaning of viscosity as such low temperatures. This is a crucial point because, as later stated at page 8, “only for ethanol elasticity theory provides nice fits to the MD data”.

Our reply:

We removed “The slightly different bubble shape for ethanol as compared to water are due to the higher viscosity of ethanol.” Because, as referee mentioned the viscosity cannot be a good parameter.

Reviewer # 1

15. [Page 8, Table II] This table is different from the one in the rebuttal letter. Please check.

Our reply:

We checked it carefully. The correct table was presented in the text.

Reviewer # 1

16. [Page 9, new figure 9] The new figure now clarifies that the sentence “hexadecane is flatter” does not mean that h_{\max}/R is lower, but that “there is a flat region on top of the hexadecane bubble”. Please adjust the text accordingly, if possible.

Our reply:

We modified figure caption as “The hexadecane forms a bubble with a larger flat region on top of the bubble in contrast to the other substances”.

Also in the text we added, page 7 first line

“... and form a bubbles. There is a flat region on top of the hexadecane bubble.”

We removed “Slightly ...”.

Reviewer # 1

17. [Page 9, last sentence] I would suggest removing “Notice”.

Our reply:

It was corrected.

Reviewer # 1

18. [Page 11, section F, line 3] possible typo: should be “obtained”. Also, this sentence is not clear.

Our reply:

It was corrected.

Reviewer # 1

19. [Page 11, section F, sentence starting with “Two RDFs are nicely fit”] Please check this sentence and remove “nice”, which may sound too colloquial.

Our reply:

We removed the word “nicely” from the sentence.

Reviewer # 1

20. [Figures 13, 16, 18] These make the layered results much clearer! If possible, however, consider that they may be printed in B/W.

Our reply:

We modified figures so that they will also OK for print in B/W.

Reviewer # 1

21. [Conclusions, general] Conclusions now start with impact, and this is good. After that, however, the reader is presented with a list of claims and the bigger picture (i.e. “we run extensive MD that allowed us to go beyond membrane theory, covering substances studied only by a few”) may get lost. It may be worth spending a little more effort to transform the conclusions in a sequential story.

Our reply:

We modified conclusion to make it clear and uniform.

Reviewer # 1

22. [Conclusions, reference [35]] I suggest substituting “our” with “a” here.

Our reply:

It was corrected.

Reviewer # 1

23. [Conclusions, paragraph starting with “the deformed graphene”] I suggest to remove “always” here, as it clashes with “except”.

Our reply:

It was corrected

Reviewer # 1

24. [Conclusions, last paragraph] remove “the important messages” and “secondly” and rephrase. Also, I am not sure that membrane theory can predict the flattening on hexadecane nanobubbles....and so I would be careful in saying that these are the best candidate for studying elasticity theory. Please check.

Our reply:

We modified the text in order to accommodate the comment from the referee, e.g. 'best candidate' is replaced by 'suitable test materials'.

Reviewer # 1

25. [Conclusions, last paragraph] In my understanding, the sentence "such scaling law is not applicable to any bubble" is more correct than inferring it does not work for bubbles below 50nm, in general.

Our reply:

We follow the suggestion made by the referee and changed the sentence accordingly.

Reply to Reviewer # 2

General comments by the reviewer:

Most of my previous concerns were addressed. However, I still feel that the conclusion from this work is not general and novel enough for Nature Communications – although I leave the judgment on these two factors to the editorial office. The writing should also be improved. The presentation should be more concise, the conclusion should be sharpened, and the number of figures should be reduced in the main text.

Our reply:

We would like to thank the referee for very careful reading of our paper and the very useful comments.

Comments:

Reviewer # 2

1. "D. Stress calculations", why use J_2 as the indication of stress?

Our reply:

We use J_2 as a scalar quantity which enabled us to color atoms with respect to their stress, see the inset of Fig. 10. The choice of this quantity was discussed earlier in Ref. 30.

We added a new reference just above Eq. (8): "[30] H. Rafii-Tabar, Phys. Rep. **4**, 235–452 (2004)".

Reviewer # 2

2. If the NaCl crystal is nucleated from solvent, would the triangular/ellipsoid/square NaCl be the favorable configurations under graphene, or the crystal will adapt the shape of graphene by forming rough surfaces?

Our reply:

The second scenario mentioned by the referee is applicable here. The crystal will adapt the shape of graphene by forming rough surfaces. We presented three theoretical-test models to show that any shape of NaCl crystal will not change its shape under high pressures. We give arguments in the text

based on the elastic properties of crystal NaCl and graphene. We did not study NaCl in a solvent environment.

We added the following line to the end of page 9

“are shown in Fig. 11(a). The crystal will adapt the shape of graphene by forming rough surfaces.”

Reply to Reviewer # 3

General comments by the reviewer:

The authors have satisfactorily addressed all my points/remarks and therefore I recommend accepting the manuscript in its present form.

Our reply:

We thank the referee for recommending the publication of our paper in Nat. Commun.